# A glutamate receptor C-tail recruits CaMKII to suppress retrograde homeostatic signaling

Sarah Perry[1,5], Yifu Han [1,5], Chengjie Qiu[1], Chun Chien [1], Pragya Goel[1], Samantha Nishimura[1], Manisha Sajnani[1], Andreas Schmid [2,4], Stephan J. Sigrist [2,3] & Dion Dickman [1] ✉

Presynaptic homeostatic plasticity (PHP) adaptively enhances neuro-transmitter release following diminished postsynaptic glutamate receptor (GluR) functionality to maintain synaptic strength. While much is known about PHP expression mechanisms, postsynaptic induction remains enigmatic. For over 20 years, diminished postsynaptic $Ca^{2+}$ influx was hypothesized to reduce CaMKII activity and enable retrograde PHP signaling at the *Drosophila* neuromuscular junction. Here, we have interrogated inductive signaling and find that active CaMKII colocalizes with and requires the GluRIIA receptor subunit. Next, we generated $Ca^{2+}$-impermeable GluRs to reveal that both CaMKII activity and PHP induction are $Ca^{2+}$-insensitive. Rather, a GluRIIA C-tail domain is necessary and sufficient to recruit active CaMKII. Finally, chimeric receptors demonstrate that the GluRIIA tail constitutively occludes retrograde homeostatic signaling by stabilizing active CaMKII. Thus, the physical loss of the GluRIIA tail is sensed, rather than reduced $Ca^{2+}$, to enable retrograde PHP signaling, highlighting a unique, $Ca^{2+}$-independent control mechanism for CaMKII in gating homeostatic plasticity.

Nervous systems are endowed with the ability to express homeostatic synaptic plasticity, a fundamental process that maintains stable functionality when confronted with internal and external perturbations. Such homeostatic control of synaptic strength occurs in the central and peripheral nervous systems of invertebrates and mammals, where adaptations in both pre- and postsynaptic compartments are observed[1–3]. One major form of homeostatic synaptic plasticity, referred to as presynaptic homeostatic potentiation (PHP), has been well-studied at the *Drosophila* neuromuscular junction (NMJ). In this system, genetic loss of the postsynaptic glutamate receptor (GluR) subunit *GluRIIA* induces a retrograde signaling system that instructs a compensatory increase in presynaptic neurotransmitter release to maintain stable levels of synaptic strength[4–6]. PHP is conserved at NMJs of rodents[7–10] and humans[11] and was recently demonstrated in the

mouse central nervous system[12]. Underscoring the importance of this process, disruption of homeostatic signaling is associated with a variety of neurological and degenerative diseases[13–16]. Important progress has been made in defining presynaptic PHP expression mechanisms[4,5,17,18] and in identifying possible retrograde signals[19,20]. However, the postsynaptic induction mechanisms that detect GluR loss and initiate retrograde PHP signaling are unknown.

When *GluRIIA* mutants were first characterized, and the phenomenon of PHP was initially described at the *Drosophila* NMJ over 20 years ago, it was hypothesized that a reduction in postsynaptic $Ca^{2+}$ influx might be the key postsynaptic signal necessary for PHP induction[21]. The possibility that reduced postsynaptic $Ca^{2+}$ is the necessary inductive signal to trigger retrograde PHP expression is an attractive idea for two reasons. First, several forms of synaptic

[1]Department of Neurobiology, University of Southern California, Los Angeles, CA, USA. [2]Institute for Biology/Genetics, Freie Universität Berlin, Takustraße 6, 14195 Berlin, Germany. [3]NeuroCure Cluster of Excellence, Charité Universitätsmedizin, Charitéplatz 1, 10117 Berlin, Germany. [4]Present address: Faculty of Life Sciences, Albstadt-Sigmaringen University, Sigmaringen, Germany. [5]These authors contributed equally: Sarah Perry, Yifu Han. ✉e-mail: dickman@usc.edu

plasticity are induced through changes in postsynaptic $Ca^{2+}$, including long-term potentiation and depression[22,23]. Second, the key trigger necessary to initiate PHP, genetic loss of postsynaptic GluRs at the fly NMJ, results in reduced postsynaptic $Ca^{2+}$ levels[24]. Postsynaptic GluRs at the fly NMJ exist as heterotetramers comprised of the common subunits GluRIIC, GluRIID, and GluRIIE plus either the GluRIIA or GluRIIB subunit[25] (Fig. 1A). A-type GluRs (referred to here as GluRA), composed of GluRIIA/C/D/E, drive the majority of synaptic currents and desensitize more slowly compared with B-type GluRs (GluRB)[26,27]. Consistent with the reduced $Ca^{2+}$ model, the only known way to induce PHP at the fly NMJ requires a loss or pharmacological blockade of GluRA receptors, which necessarily also diminishes postsynaptic $Ca^{2+}$ levels. However, this model has not been directly tested because the coupling between diminished GluRA abundance and reduced postsynaptic $Ca^{2+}$ has never been separated.

There is compelling evidence to support the hypothesis that reduced $Ca^{2+}$ triggers a reduction in postsynaptic Calmodulin-

dependent Kinase II (CaMKII) activity to enable retrograde PHP signaling at the *Drosophila* NMJ. First, postsynaptic overexpression of a constitutively active, phosphomimetic *CaMKII^{T287D}* is capable of blocking PHP expression in *GluRIIA* mutants[28,29]. In addition, reduced levels of T287 phosphorylation (autoactivated; pCaMKII) were observed at postsynaptic compartments in *GluRIIA* mutants[24,29,30]. CaMKII is an appealing potential sensor of reduced $Ca^{2+}$ in postsynaptic compartments at the fly NMJ. This enzyme functions as a central postsynaptic signaling node to detect and respond to changes in $Ca^{2+}$ during the induction of synaptic plasticity[22,31]. CaMKII forms a unique 12-mer holoenzyme that is immensely abundant in the nervous system[32]. The inactive CaMKII holoenzyme exists as a compact structure, inhibiting access to substrates (Fig. 1D). Upon a rise in $Ca^{2+}$, the regulatory region of CaMKII is displaced, which enables autophosphorylation of a key Thr residue (T287 in *Drosophila*; T286 in mammals) by neighboring subunits in the holoenzyme. CaMKII can persist in this activated, "autonomous" state long after the transient change in

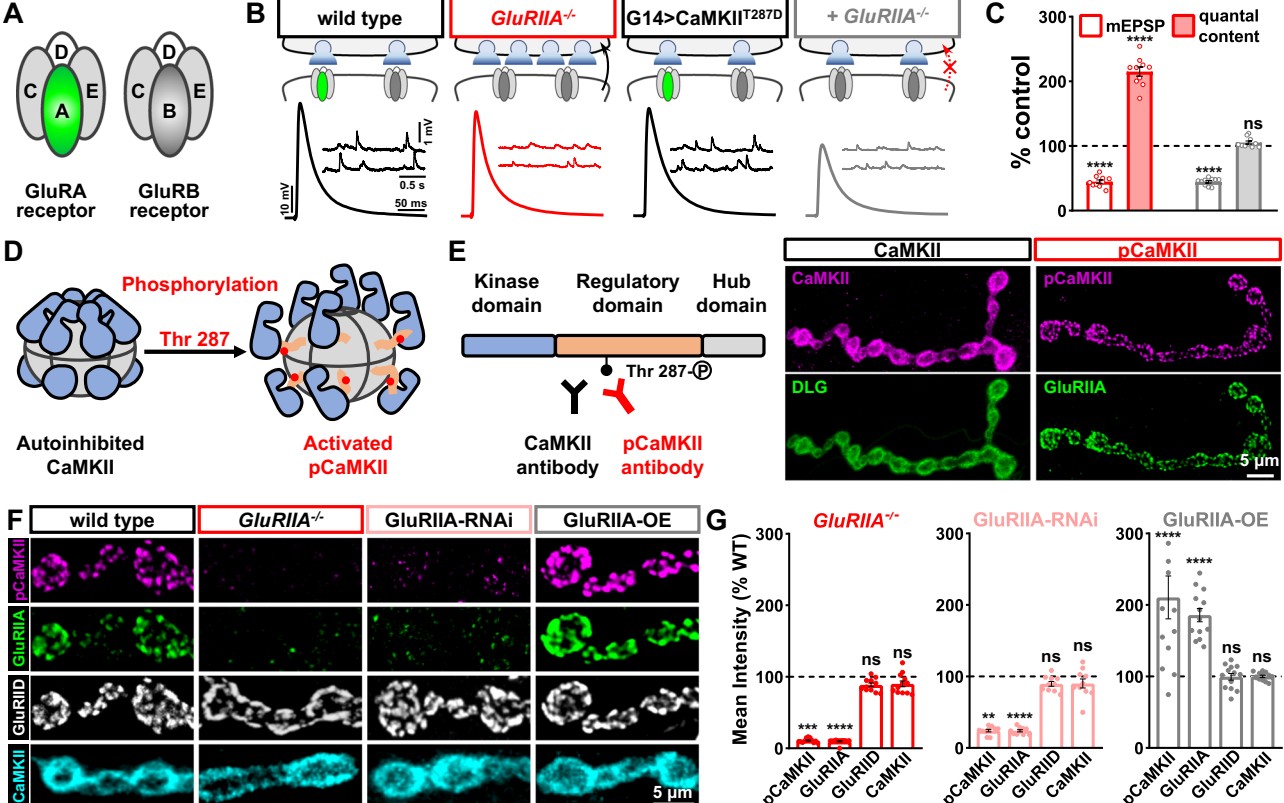

**Fig. 1 | Active CaMKII localizes to postsynaptic glutamate receptor fields and correlates with GluRIIA expression. A** Schematic depicting the subunit composition of GluRA and GluRB glutamate receptor subtypes at the *Drosophila* NMJ. **B** Schematic and representative traces illustrate that postsynaptic expression of constitutively active *CaMKII* blocks the chronic expression of presynaptic homeostatic potentiation (PHP) that is normally induced by loss of the *GluRIIA* subunit. Genotypes: wild type (*w^{1118}*); *GluRIIA^{-/-}* (*w;GluRIIA^{SP16}*); G14 > CaMKII^{T287D} (*w;G14-GAL4/+;UAS-CaMKII^{T287D}/+*); G14 > CaMKII^{T287D} + *GluRIIA^{-/-}* (*w;G14-GAL4,GluRIIA^{SP16}/GluRIIA^{SP16};UAS-CaMKII^{T287D}/+*). (**C**) Quantification of mEPSP amplitude and quantal content values in the indicated genotypes normalized to baseline values (wild type: $n = 10$; GluRIIA^{-/-}, $n = 10$, $p < 0.0001$ for mEPSP, $p < 0.0001$ for QC; G14 > CaMKII^{T287D}: $n = 10$; G14 > CaMKII^{T287D} + GluRIIA^{-/-}: $n = 10$, $p < 0.0001$ for mEPSP, $p = 0.072$ for QC, unpaired, two-tailed *t*-test, with a significance value of 0.05). **D** Schematic of inactive and active versions of the CaMKII holoenzyme through phosphorylation of Thr-287. **E** CaMKII protein structure and the antigenic sites used to generate anti-CaMKII- and -pCaMKII- antibodies. Representative confocal images of the muscle 4 NMJ immunostained with anti-CaMKII or anti-pCaMKII antibodies

and co-stained with either the postsynaptic scaffold Disks Large (DLG) or GluRIIA. **F** Representative images of NMJ boutons immunostained with anti-pCaMKII, -GluRIIA, -GluRIID, and -CaMKII in the indicated genotypes: GluRIIA-RNAi (*w;G14-Gal4/+;UAS-GluRIIA^{RNAi}/+*); GluRIIA-OE (*w;+;MHC-GluRIIA*). **G** Quantification of mean fluorescence intensity values of pCaMKII, GluRIIA, GluRIID, and CaMKII normalized to wild-type values in the indicated genotypes (wild type: $n = 13$; GluRIIA^{-/-}, $n = 12$, $p < 0.0001$ for GluRIIA, $p = 0.009$ for pCaMKII, $p = 0.9996$ for GluRIID, $p = 0.1497$ for CaMKII; GluRIIA-RNAi: n = 10, $p < 0.0001$ for GluRIIA, $p = 0.0080$ for pCaMKII, p = 0.2027 for GluRIID, p = 0.1776 for CaMKII; GluRIIA-OE: $n = 13$, $p < 0.0001$ for GluRIIA, $p < 0.0001$ for pCaMKII, $p = 0.9996$ for GluRIID, $p > 0.9999$ for CaMKII). Repeated measures One-way ANOVA with Dunnett's multiple comparisons test with a significance value of 0.05. *p* Value adjusted for multiple comparisons. Error bars indicate ±SEM. Asterisks indicate statistical significance using *t*-test or one-way ANOVA: **$p < 0.01$, ***$p < 0.001$, ****$p < 0.0001$; ns, not significant. *n* values indicate biologically independent NMJs. Absolute values for normalized data are summarized in Table S1. **E, F** Experiments were repeated four times independently with similar results. Source data are provided as a Source Data file.

$Ca^{2+}$ [23,32]). While not an absolute measure of CaMKII enzymatic activity, the phosphorylation state of the T287 residue is an accurate indicator of the "open" conformation of the CaMKII holoenzyme, which correlates well with activity levels [23,31]. Interestingly, there is also evidence that the C-tail of GluRs and other scaffolds can activate CaMKII independently of $Ca^{2+}$ [23]. The ability of CaMKII to serve as a secondary $Ca^{2+}$ sensor as well as an integral component of the postsynaptic apparatus further reinforces the potential of this enzyme to transform transient changes in activity to long term adaptations in synaptic function. However, the role of CaMKII in PHP signaling, and even whether reduced CaMKII activity is necessary, remains enigmatic.

We have developed a host of specific antibodies and new mutant alleles using CRISPR/Cas9 approaches to interrogate the role of *GluRIIA*, postsynaptic $Ca^{2+}$, and CaMKII activity in retrograde PHP induction. This work has revealed that CaMKII activity and PHP induction is not influenced by diminished $Ca^{2+}$ at postsynaptic compartments. Rather, active CaMKII requires an intimate interaction with the GluRIIA C-tail. Loss of this interaction is necessary to allow retrograde signaling and PHP expression, highlighting a unique and unanticipated inductive mechanism.

## Results

### Active CaMKII co-localizes and correlates with GluRIIA expression

To investigate postsynaptic CaMKII function in retrograde PHP signaling, we first replicated the experiment that most clearly established a relationship between CaMKII and PHP expression [28]. At *Drosophila* NMJs, genetic deletion of the *GluRIIA* subunit leads to loss of GluRA receptors and a reduction in mEPSP amplitude, as expected [21]. However, evoked EPSP amplitudes remain similar to wild-type levels due to a homeostatic increase in presynaptic neurotransmitter release (quantal content), indicating PHP expression (Fig. 1B, C). When constitutively active CaMKII ($CaMKII^{T287D}$) is postsynaptically overexpressed in *GluRIIA* mutants, retrograde homeostatic signaling is blocked, with no increase in presynaptic release observed (Fig. 1B, C). This provides evidence that a reduction in CaMKII activity may be necessary to allow retrograde PHP signaling.

In *GluRIIA* mutants, reductions in pCaMKII immunofluorescence levels have been observed at the NMJ [24,29,30]. However, because the commercial antibodies used in these studies were developed against rodent pCaMKII antigens, it is not clear that these antibodies reflect specific levels and localization of *Drosophila* CaMKII. Thus, we generated new *Drosophila*-specific CaMKII antibodies using peptides containing the *Drosophila* CaMKII regulatory domain (Fig. 1E). We successfully developed two highly specific antibodies: one antibody recognizes total CaMKII levels (anti-CaMKII), while the other recognizes only the active (T287-phosphorylated) form of the enzyme (anti-pCaMKII; Fig. 1E). We performed several experiments to validate the specificity of the CaMKII antibodies we generated. First, we manipulated postsynaptic CaMKII expression levels and activity at the *Drosophila* NMJ, finding that pCaMKII and CaMKII mRNA levels and immunofluorescence intensities changed in the expected ways with CaMKII expression or activity (Figs. S1 and S2). Second, immunoblot analysis of anti-CaMKII on muscle lysates revealed a single band at the expected molecular mass (~58 kDa) that was reduced with CaMKII-RNAi and enhanced with overexpression (Fig. S2).

Remarkably, these new CaMKII antibodies revealed striking differences in CaMKII localization and activity at the *Drosophila* NMJ. Total CaMKII localized to postsynaptic compartments, exhibiting a high degree of overlap with the postsynaptic density marker Disks Large (DLG; Fig. 1E). In contrast, pCaMKII showed a punctate distribution at postsynaptic areas of the NMJ, co-localizing with the GluRIIA subunit (Fig. 1E). Next, we examined CaMKII and pCaMKII levels in *GluRIIA* mutants. In previous studies using commercial (mammalian) CaMKII antibodies, pCaMKII appeared diffuse at

postsynaptic compartments at the fly NMJ and was reduced by ~50% in *GluRIIA* mutants [24,29,30]. However, using the new antibodies, we observed no change in total CaMKII staining, while, unexpectedly, pCaMKII signals were entirely absent in *GluRIIA* mutants (Fig. 1F, G). Conversely, pCaMKII levels were increased when *GluRIIA* was overexpressed, while no change was observed in total CaMKII (Fig. 1F, G). Although anti-CaMKII and anti-DLG exhibited similar immunostaining patterns, we confirmed that anti-CaMKII did not cross-react with DLG (Fig. S1C, D and Fig. S2A). Together, these data provide evidence for an unanticipated tight coupling between GluRIIA levels and CaMKII activity.

Finally, we tested whether manipulation of CaMKII may reciprocally control GluRIIA receptor levels, as well as exploring the relationship between CaMKII and pCaMKII. We observed significant reductions in pCaMKII staining when we knocked down CaMKII expression or overexpressed inhibitory CaMKII peptides (Fig. S1A, B). However, GluRIIA and GluRIID levels were only modestly impacted in these conditions (Fig. S1A, B). We also overexpressed wild-type *CaMKII* and constitutively active $CaMKII^{T287D}$. Both manipulations resulted in marked increases in pCaMKII staining. Again, GluRIIA and GluRIID levels were only modestly affected in these genotypes. On the other hand, CaMKII staining is reduced when CaMKII is knocked down and unaffected by inhibitory peptide expression (Fig. S1C, D). Interestingly, CaMKII staining is also reduced when $CaMKII^{T287D}$ is overexpressed, which may indicate an equilibrium between active and total CaMKII protein. Thus, while CaMKII activity or levels do not reciprocally regulate GluRIIA levels, CaMKII activity, as indicated by T287 phosphorylation, is sensitively tuned to the abundance of GluRIIA.

### pCaMKII levels are insensitive to reductions in and even elimination of postsynaptic $Ca^{2+}$

Active pCaMKII is apparently tightly linked to GluRIIA expression. We considered two possibilities to explain the relationship between pCaMKII and GluRIIA. First, since GluRA receptors drive the majority of synaptic currents and $Ca^{2+}$ influx [26,27], the gain or loss of these receptors will have a major impact on postsynaptic $Ca^{2+}$ levels. Given that synaptic $Ca^{2+}$ levels are well established to be capable of influencing CaMKII activity, and stimulating T287 autophosphorylation [23,32], postsynaptic $Ca^{2+}$ levels at the fly NMJ may therefore tune the levels of active pCaMKII (schematized in Fig. 2A). In the next series of experiments, we tested whether CaMKII activity is sensitive to postsynaptic $Ca^{2+}$.

We first attempted to disrupt ionic influx through postsynaptic GluRA receptors using a previously developed *GluRIIA* transgene. This transgenic *GluRIIA* allele, $UAS\text{-}GluRIIA^{M614R}$, was designed to disrupt ionic influx through GluRA receptors by presumably acting as a "dominant negative", antagonizing endogenous GluRA receptors [27] (Fig. S3A). However, while we did observe a reduction in mEPSP amplitude similar to previous reports (Table S1), GluR staining revealed poor receptor trafficking, with reductions in GluRIIA, GluRIIB, and GluRIID levels at the NMJ (Fig. S3B, C). We also observed high levels of GluRIIA that accumulated in intracellular compartments throughout the muscle, indicating that postsynaptic overexpression of this transgene generally disrupts GluR trafficking (Fig. S3B). Thus, postsynaptic overexpression of the $GluRIIA^{M614R}$ allele induced a general GluR knockdown, rendering it an ineffective method for determining whether postsynaptic $Ca^{2+}$ impacts pCaMKII levels, independently of GluRIIA abundance.

Therefore, we developed two new approaches to selectively reduce postsynaptic $Ca^{2+}$ levels at the larval NMJ without disrupting GluR abundance. First, we used CRISPR/Cas9 gene editing to generate $Ca^{2+}$ impermeable GluRA receptors, while still allowing other ionic conductances. This was accomplished by mutating a single amino acid in the selectivity pore in the endogenous *GluRIIA* locus (Fig. 2B). AMPA and kainate-type GluRs that are $Ca^{2+}$ permeable contain a glutamine

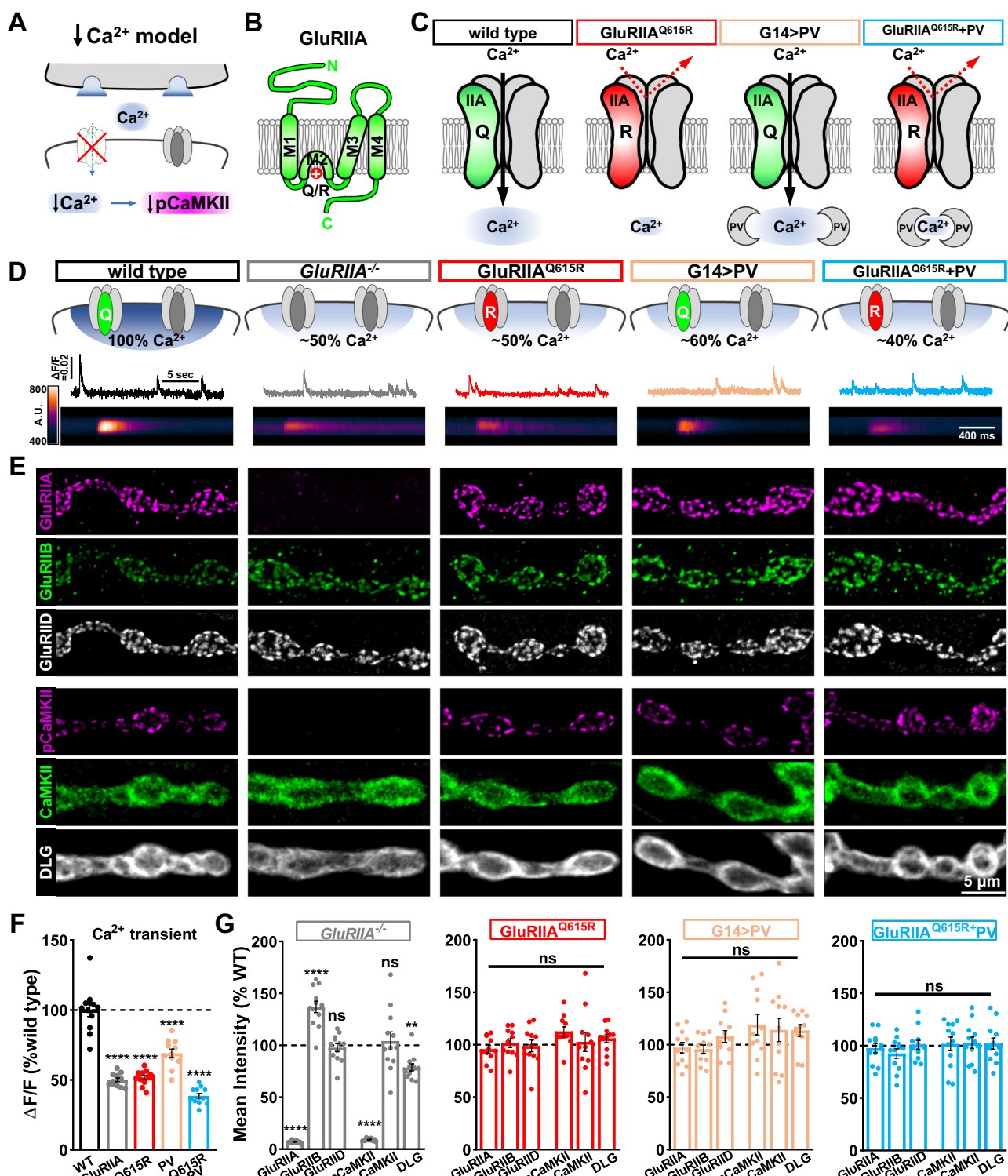

(Q) residue in the M2 domain; some GluR subunits, including mammalian AMPA, and *Drosophila* kainate GluRs are unable to conduct $Ca^{2+}$ when this Q is changed to the positively charged arginine (R) amino acid[33–36] (Fig. 2B). We targeted the orthologous amino acid in GluRIIA (Q615) for mutagenesis at the endogenous locus to generate a *GluRIIA[Q61SR]* allele (Fig. 2B, C). We quantified postsynaptic $Ca^{2+}$ levels using GCaMP6f targeted to postsynaptic NMJs (SynapGCaMP6f[24]) and quantified quantal $Ca^{2+}$ events (Fig. 2D). Quantal signals in *GluRIIA* mutants were reduced by ~50% compared to wild type as observed previously[24], consistent with a major reduction in postsynaptic $Ca^{2+}$

due to loss of GluRAs (Fig. 2D, F). Importantly, while GluRA and GluRB levels were unchanged in *GluRIIA[Q61SR]* mutants (Fig. 2E, G), a similar ~50% reduction in postsynaptic $Ca^{2+}$ was observed that was statistically indistinguishable from *GluRIIA* null mutants (Fig. 2D, F). This demonstrates that the *GluRIIA[Q61SR]* allele reduces postsynaptic $Ca^{2+}$ influx to the same levels found in *GluRIIA* mutants without altering postsynaptic GluR abundance. Finally, we assayed pCaMKII levels in *GluRIIA[Q61SR]* mutants and found no significant difference in either total CaMKII or, importantly, pCaMKII levels (Fig. 2E, G). Thus, $Ca^{2+}$ influx through GluRA receptors does not modulate CaMKII activity.

**Fig. 2 | pCaMKII levels are insensitive to reductions in postsynaptic Ca²⁺.**
**A** Schematic illustrating that loss of GluRA receptors decreases postsynaptic $Ca^{2+}$ influx and may reduce pCaMKII levels at postsynaptic compartments. **B** Membrane topology of the GluRIIA subunit with the Q615R mutation shown in the pore-forming M2 domain. **C** Schematics illustrating $Ca^{2+}$ permeability through GluRA receptors and the $Ca^{2+}$ buffer parvalbumin (PV), with the associated reductions in $Ca^{2+}$ observed in postsynaptic compartments. Genotypes: *GluRIIA^Q615R* (*w;GluR-IIA^Q615R*); G14 > PV (*w;G14-GAL4/+;UAS-PV/+*); *GluRIIA^Q615R* + G14 > PV (*w;GluR-IIA^Q615R,G14-GAL4/GluRIIA^Q615R;UAS-PV/+*). **D** Schematized GluRs and postsynaptic $Ca^{2+}$ levels with representative $Ca^{2+}$ imaging traces. Line scans below are derived from postsynaptic GCaMP6f images of individual spontaneous $Ca^{2+}$ transients in the indicated genotypes: wild type (*w;SynapGCaMP6f/+*), *GluRIIA^−/−* (*w;GluRIIA^SP16;SynapGCaMP6f/+*), *GluRIIA^Q615R* (*w;GluRIIA^Q615R;SynapGCaMP6f/+*), G14 > PV (*w;G14-GAL4/+;UAS-PV/SynapGCaMP6f*), *GluRIIA^Q615R* + G14 > PV (*w;GluR-IIA^Q615R,G14-GAL4/GluRIIA^Q615R;UAS-PV/SynapGCaMP6f*). **E** Representative images of NMJ boutons immunostained with anti-GluRIIA, -GluRIIB, -GluRIID, -pCaMKII, -CaMKII, and -DLG antibodies in the indicated genotypes shown in (**D**) without *SynapGCaMP6f* expression. Experiments were repeated four times independently with similar results. **F** Quantification of the normalized changes in fluorescence intensity (Δ*F/F*) of spontaneous $Ca^{2+}$ transient events at individual boutons in the indicated genotypes in (**D**) (wild type: $n = 12$; GluRIIA^−/−, $n = 12$, $p < 0.0001$; GluR^Q615R:

$n = 12$, $p < 0.0001$; G14 > PV: $n = 11$, $p < 0.0001$; GluRIIA^Q615R + G14 > PV: $n = 12$, $p < 0.0001$). Repeated measures one-way ANOVA with Dunnett's multiple comparisons test with a significance value of 0.05. $p$ Value adjusted for multiple comparisons. **G** Quantification of the mean fluorescence intensity of anti-GluRIIA, -GluRIIB, -GluRIID, -pCaMKII, -CaMKII, and -DLG in the indicated genotypes normalized to wild-type values. (wild type: $n = 18$; GluRIIA^−/−, $n = 12$, $p < 0.0001$ for GluRIIA, $p < 0.0001$ for GluRIIB, $p = 0.9490$ for GluRIID, $p < 0.0001$ for pCaMKII, $p = 0.9848$ for CaMKII, $p = 0.0010$ for DLG; GluRIIA^Q615R: $n = 12$, $p = 0.6473$ for GluRIIA, $p > 0.9999$ for GluRIIB, $p = 0.9889$ for GluRIID, $p = 0.3169$ for pCaMKII, $p = 0.9971$ for CaMKII, $p = 0.6843$ for DLG; G14 > PV: $n = 11$, $p = 0.8479$ for GluRIIA, $p = 0.7851$ for GluRIIB, $p = 0.6389$ for GluRIID, $p = 0.0547$ for pCaMKII, $p = 0.4991$ for CaMKII, $p = 0.0616$ for DLG; GluRIIA^Q615R + G14 > PV: $n = 12$, $p = 0.8829$ for GluRIIA, $p = 0.4087$ for GluRIIB, $p > 0.9999$ for GluRIID, $p = 0.9980$ for pCaMKII, $p = 0.9965$ for CaMKII, $p = 0.9903$ for DLG). Repeated measures One-way ANOVA with Dunnett's multiple comparisons test with a significance value of 0.05. $p$ Value adjusted for multiple comparisons. Error bars indicate ±SEM. Asterisks indicate statistical significance using One-way ANOVA: *$p < 0.05$, **$p < 0.01$, ****$p < 0.0001$; ns, not significant. $n$ values indicate biologically independent NMJs. Absolute values for normalized data are summarized in Table S1. Source data are provided as a Source Data file.

---

To determine if intracellular $Ca^{2+}$ in postsynaptic compartments, independent of conductance through GluRA receptors, was necessary to maintain pCaMKII activity, we cloned the mammalian $Ca^{2+}$ buffer *parvalbumin* (PV) into the strong expression vector pACU2[37]. PV localized to postsynaptic NMJ compartments when expressed in muscle (Fig. S4), and we observed no significant change in GluR levels (Fig. 2E, G). However, quantal $Ca^{2+}$ imaging revealed an ~40% reduction, reducing $Ca^{2+}$ to levels close to that observed in *GluRIIA* mutants. Consistent with the results for *GluRIIA^Q16R*, we found no significant difference in total CaMKII or pCaMKII levels in this condition (Fig. 2E, G). To reduce postsynaptic $Ca^{2+}$ levels below that observed in *GluRIIA* mutants alone, we combined the *GluRIIA^Q16R* allele with postsynaptic PV overexpression. Postsynaptic $Ca^{2+}$ was reduced by over 60% compared to the wild type (Fig. 2D, F), while GluRIIA, GluRIID, CaMKII, and pCaMKII levels were not significantly different from the wild type (Fig. 2E, G). We also did not observe differences in postsynaptic $Ca^{2+}$ responses driven by endogenous patterns of motor neuron firing at *GluRIIA* NMJs compared to wild type (Fig. S5A, B), and pCaMKII levels were only modestly reduced after 30 min incubation in $Ca^{2+}$ chelator BAPTA-AM (Fig. S5C, D). Together, this indicates that CaMKII activity is insensitive to reductions in synaptic $Ca^{2+}$ at postsynaptic compartments.

Finally, we asked whether CaMKII activity is sensitive to synaptic transmission itself. Synaptic activity at mammalian glutamatergic synapses is well known to activate CaMKII through two cooperative mechanisms: $Ca^{2+}$ influx mediated by the opening of postsynaptic GluRs, and conformational changes in the C-tail of NMDARs driven by glutamate binding[23]. To silence synaptic activity at NMJs, we developed a botulinum neurotoxin (BoNT-C) that targets the SNARE component Syntaxin for cleavage and blocks all synaptic vesicle release[38,39]. Expression of this toxin using a driver that targets a subset of motor neurons (*OK319-GAL4*) eliminates all miniature and evoked neurotransmission (Fig. 3A). To confirm the absence of any $Ca^{2+}$ influx at postsynaptic compartments silenced by BoNT-C expression, we generated a new postsynaptic GCaMP indicator with improved speed and sensitivity based on SynapGCaMP6f[24], where GCaMP6f was replaced with GCaMP8f[40] to make SynapGCaMP8f[39]. We confirmed a complete absence of postsynaptic $Ca^{2+}$ activity at NMJs silenced by BoNT-C expression, as expected due to the elimination of glutamate release. Importantly, pCaMKII levels remained unchanged at synapses chronically silenced throughout development by BoNT-C, while pCaMKII was abolished at *GluRIIA* mutant NMJs silenced by BoNT-C (Fig. 3C, D). Together, this suggests that at the *Drosophila* NMJ, CaMKII activity is regulated not by synaptic glutamate and/or $Ca^{2+}$ influx, but

rather through the physical presence of the kainate receptor subunit GluRIIA.

### Reduced postsynaptic Ca²⁺ levels are not sufficient to induce PHP expression

When *GluRIIA* mutants were first characterized over 20 years ago, it was immediately speculated that reduced $Ca^{2+}$ influx due to loss of high conductance GluRA receptors may be the primary signal necessary to induce retrograde PHP signaling[21]. Since this seminal study, this idea has been consistently invoked in subsequent studies[24,28,41]. However, this hypothesis has not been directly tested. We, therefore, assessed synaptic function in conditions in which postsynaptic $Ca^{2+}$ levels are diminished to the same extent as found in *GluRIIA* null mutants (*GluRIIA^Q615R* mutants and PV overexpression) and even further reduced below this state (*GluRIIA^Q615R* + G14 > PV). If reduced postsynaptic $Ca^{2+}$, as observed in *GluRIIA* mutants, is the key inductive signal for retrograde signaling and PHP expression, then one should expect synaptic strength (EPSP amplitude) and quantal content to be enhanced in the manipulations that reduce postsynaptic $Ca^{2+}$, while miniature activity remains unchanged from baseline. Electrophysiological recordings from *GluRIIA* mutants show mEPSP amplitudes reduced over 50% compared with wild type, as expected, but similar EPSP amplitude due to a homeostatic increase in presynaptic release (quantal content; Fig. 4A–E). It is this increase in quantal content that defines PHP expression. Recordings from *GluRIIA^Q615R*, PV overexpression, and *GluRIIA^Q615R* + PV overexpression NMJs revealed mEPSP amplitudes unchanged from the wild type, as expected. However, no significant difference in EPSP amplitude or quantal content was found (Fig. 4A–E). This indicates that despite reduced postsynaptic $Ca^{2+}$ levels comparable to or even below that observed in *GluRIIA* mutants, no change in presynaptic neurotransmitter release is observed. Therefore, reduced postsynaptic $Ca^{2+}$ influx alone is insufficient to induce retrograde PHP signaling.

### Truncation of the GluRIIA C-tail prevents activation of postsynaptic CaMKII

Having ruled out the conventional $Ca^{2+}$ influx model for controlling pCaMKII activity and PHP induction, we next tested an alternative model in which pCaMKII is stabilized directly or indirectly through a biochemical interaction with a GluR C-tail (schematized in Fig. 5A). Mammalian NMDARs recruit and activate CaMKII directly through binding sites encoded in their C-terminal cytosolic tails[42]. The GluN2B C-tail contains a "GluN2B-tide" region that is capable of recruiting CaMKII and promoting T286 phosphorylation[23,42]. Other protein

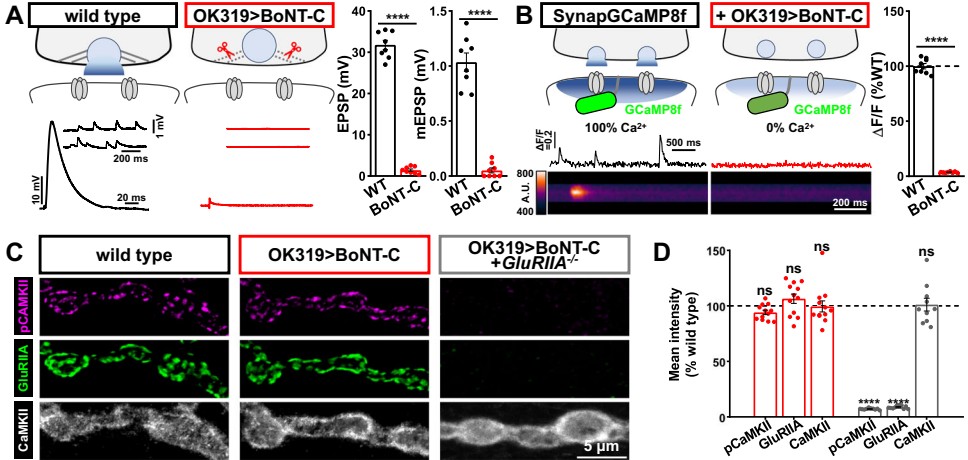

**Fig. 3 | CaMKII activity remains unchanged when synaptic transmission is silenced by BoNT-C. A** Schematic depicting cleavage of the synaptic vesicle SNARE complex by BONT-C expression (*w;OK319-Gal4/+;UAS-BoNT-C/+*), preventing all vesicle fusion. Representative electrophysiological traces and quantifications are shown for the indicated genotype, confirming no evoked or miniature transmission at NMJs silenced by BoNT-C. Quantification of average mEPSP amplitude (**B**), EPSP amplitude (**C**), and quantal content (**D**) values in the indicated genotypes (wild type: *n* = 8; OK319 > BoNT-C, *n* = 8, *p* < 0.0001 for mEPSP, *p* < 0.0001 for EPSP, unpaired, two-tailed *t*-test). **B** Schematic and postsynaptic Ca²⁺ imaging using a new SynapGCaMP8f indicator in wild type (*w;SynapGCaMP8f/+*) and BoNT-C silenced (+OK319 > BoNT-C: *w;OK319-GAL4/+;SynapGCaMP8f/UAS-BoNT-C*) NMJs. Quantification of transients, confirming no synaptic Ca²⁺ events in postsynaptic compartments following BoNT-C silencing. (wild type: *n* = 8; OK319 > BoNT-C, *n* = 8, *p* < 0.0001, unpaired, two-tailed *t*-test, with a significance value of 0.05). **C** Representative images of NMJs immunostained for pCaMKII, GluRIIA, and CaMKII

in wild type, BoNT-C silenced, and *GluRIIA* mutants with BoNT-C silencing (*w;OK319-GAL4,GluRIIA^SP16^/GluRIIA^SP16^;UAS-BoNT-C/+*). Experiments were repeated three times independently with similar results. **D** Quantification of mean fluorescence intensity of the indicated antibodies (wild type: *n* = 12; OK319 > BoNT-C, *n* = 12, *p* = 0.1373 for pCaMKII, *p* = 0.3770 for GluRIIA, *p* = 0.9974 for CaMKII; OK319 > BoNT-C + GluRIIA⁻/⁻: *n* = 10, *p* < 0.0001 for pCaMKII, *p* < 0.0001 for GluR-IIA, *p* = 0.9835 for CaMKII). Repeated measures one-way ANOVA with Dunnett's multiple comparisons test with a significance value of 0.05. *p* Value adjusted for multiple comparisons. Note that pCaMKII levels are unchanged in BoNT-C silenced NMJs, while pCaMKII is abolished by loss of *GluRIIA*. Error bars indicate ±SEM. Asterisks indicate statistical significance using *t*-test or one-way ANOVA: ****\*p* < 0.0001; ns, not significant. *n* values indicate biologically independent NMJs. Absolute values for normalized data are summarized in Table S1. Source data are provided as a Source Data file.

domains can also serve as scaffolds to bind and activate CaMKII, including the *Drosophila* potassium channel EAG[43,44]. Inspired by these studies, we hypothesized that part of the GluRIIA C-tail may function similarly to promote CaMKII recruitment, stabilization, and/or activation. We observed a region in the GluRIIA C-tail with homology to both the GluN2B C-tail domain, the CaMKII autoinhibitory domain, and a region in the *Drosophila* potassium channel Eag (Fig. 5B); this suggested potential interaction sequences with CaMKII. In particular, a kinase consensus sequence (R-Q/R-X-T/S-X-D/E) located at the distal end of the GluRIIA C-tail could, in principle, interact with CaMKII in a similar manner. This terminal region of the GluRIIA C-tail was therefore an attractive target to potentially interact with CaMKII.

Although the ability of the NMDA receptor C-tail to bind CaMKII and promote "autonomous" activity is well established in mammalian systems, GluRs at the *Drosophila* NMJ are kainate-type receptors[33], which, to our knowledge, have never been shown to directly bind and promote CaMKII activity. There are two NMDA receptors encoded in the *Drosophila* genome, NMDAR1, and NMDAR2, and it is possible that one or both receptors actually interact with muscle CaMKII and modulate its activity. We, therefore, mapped expression and generated null mutations of *NMDAR1* and *NMDA2*. First, we find that both receptors are expressed in the larval brain and in motor neurons but are not expressed in muscle (Fig. S6B). We generated null mutations in both receptors[45] (Fig. S6A) and found no defects in baseline electrophysiological function or PHP expression in these mutants (Fig. S6C, D). Finally, we found no changes in pCaMKII or CaMKII immunostaining at mutant NMJs of either receptor compared to the wild type (Fig. S6E, F). Thus, NMDA-type GluRs are not expressed and do not function in larval muscles to control CaMKII activity or gate retrograde PHP signaling.

To specifically test whether the GluRIIA C-tail subserves a putative function in promoting pCaMKII localization at postsynaptic GluR receptive fields and CaMKII activity, we used CRISPR/Cas9

mutagenesis to truncate the C-terminal tail of the *GluRIIA* subunit at the endogenous locus (Fig. 5C). Specifically, we designed two guide RNAs (sgRNAs) for Cas9 mutagenesis targeting the final 19 codons in the terminal *GluRIIA* exon (Fig. 5C). This approach generated two independent truncation alleles (*GluRIIA^ΔC20^* and *GluRIIA^ΔC6^*) that disrupted the last 20 and 6 amino acids of the GluRIIA C-tail, respectively. We also generated a C-tail deletion in the GluRIIA^Q615R^ allele, which ablated the final 19 amino acids (*GluRIIA^QRΔC19^*). GluR staining in these alleles confirmed that GluRA receptors trafficked normally, with no significant differences observed in GluRIIA or GluRIID levels compared to the wild type (Fig. 5D, E). The antigen of the monoclonal GluRIIA antibody 8B4D2 is unknown, but we confirmed that it is in the extracellular region of GluRIIA (Fig. S7). To confirm that the C-tail was indeed disrupted in these new *GluRIIA* alleles, we generated an antibody against the terminal 18 amino acids of the GluRIIA C-tail (anti-GluRIIA^tail^; Fig. 5C) and validated the antigen was intracellular at the fly NMJ (Fig. S5). Using the GluRIIA^tail^ antibody, we confirmed that the GluRIIA C-tail was disrupted in each of the new *GluRIIA* truncation alleles, as expected (Fig. 5D, E). Remarkably, pCaMKII was not detectable in either *GluRIIA^ΔC20^*, *GluRIIA^ΔC6^*, or *GluRIIA^QRΔC19^*, while total CaMKII levels were unchanged (Fig. 5D, E). Thus, a short sequence at the C-terminal cytosolic tail of the GluRIIA subunit is necessary for activated pCaMKII to be present at postsynaptic NMJ compartments, consistent with this region serving as a CaMKII docking and activation site in a kainate-type receptor, analogous to NMDARs at mammalian central synapses.

**Loss of pCaMKII does not induce retrograde homeostatic signaling**

Active pCaMKII is lost in *GluRIIA* mutants, and postsynaptic overexpression of constitutively active *CaMKII* blocks the expression of PHP (Fig. 1). We, therefore, considered the possibility that the absence of active pCaMKII at the NMJ may be sufficient to enable retrograde

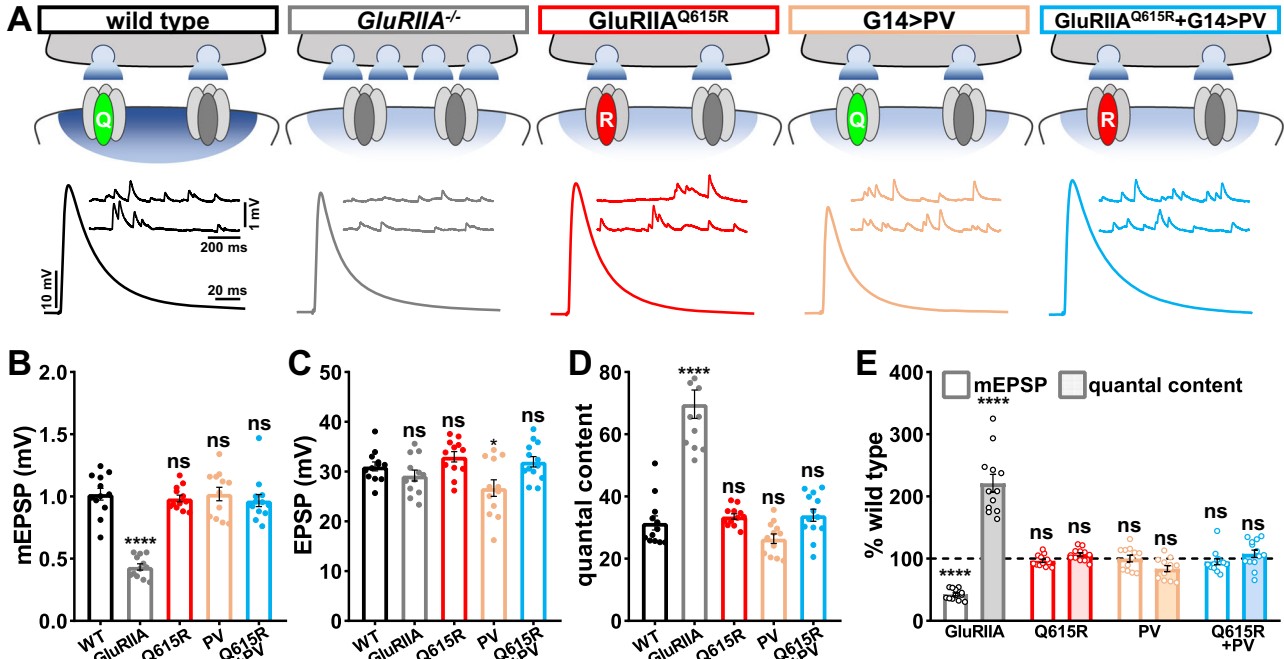

**Fig. 4 | Reductions in postsynaptic Ca²⁺ levels do not induce PHP expression. A** Schematic depicting genetic manipulations that reduce postsynaptic Ca²⁺ levels at the *Drosophila* NMJ. Representative electrophysiological traces are shown below each indicated genotype. **B–D** Quantification of average mEPSP amplitude (**B**), EPSP amplitude (**C**), and quantal content (**D**) values in the indicated genotypes in (**A**) (wild type: $n = 12$; GluRIIA⁻/⁻, $n = 12$, $p < 0.0001$ for mEPSP, $p = 0.607$ for EPSP, $p < 0.0001$ for QC; GluR^Q615R: $n = 12$, $p = 0.946$ for mEPSP, $p = 0.607$ for EPSP, $p = 0.942$ for QC; G14 > PV: $n = 12$, $p = 0.999$ for mEPSP, $p = 0.041$ for EPSP, $p = 0.437$ for QC; GluRIIA^Q615R + G14 > PV: $n = 13$, $p = 0.812$ for mEPSP, $p = 0.942$ for EPSP, $p = 0.893$ for QC). Repeated measures one-way ANOVA with Dunnett's multiple comparisons test with a significance value of 0.05. *p* Value adjusted for multiple comparisons. **E** Quantification of mEPSP and quantal content values of the indicated genotypes normalized to wild-type values (wild type: $n = 12$; GluRIIA⁻/⁻, $n = 12$, $p < 0.0001$ for mEPSP, $p < 0.0001$ for QC; GluR^Q615R: $n = 12$, $p = 0.946$ for mEPSP, $p = 0.942$ for QC; G14 > PV: $n = 12$, $p = 0.999$ for mEPSP, $p = 0.437$ for QC; GluRIIA^Q615R + G14 > PV: $n = 13$, $p = 0.812$ for mEPSP, $p = 0.893$ for QC). Repeated measures One-way ANOVA with Dunnett's multiple comparisons test with a significance value of 0.05. *p* Value adjusted for multiple comparisons. Error bars indicate ±SEM. Asterisks indicate statistical significance using one-way ANOVA: ****$p < 0.0001$; ns, not significant. *n* values indicate biologically independent NMJs. Absolute values for normalized data are summarized in Table S1. Source data are provided as a Source Data file.

PHP signaling alone, or perhaps in combination with reduced Ca²⁺ influx. Thus, we performed electrophysiology in the *GluRIIA* C-tail truncation alleles and assessed whether any change in presynaptic neurotransmitter release was observed. mEPSP amplitude was reduced in *GluRIIA* null mutants, while mEPSP amplitude was unchanged compared to wild type in each of the new *GluRIIA* C-tail truncation mutants, as expected (Fig. 6A, B). However, while presynaptic neurotransmitter release was nearly doubled in *GluRIIA* null mutants, no change in EPSP amplitude or quantal content was found in any of the *GluRIIA* C-tail truncation alleles (Fig. 6A–E). Importantly, no change in quantal content indicative of PHP expression was found even when loss of pCaMKII was combined with diminished Ca²⁺ influx in the *GluRIIA^QRΔC19* allele (Fig. 6A–E). Thus, loss of pCaMKII, even in combination with reduced Ca²⁺ influx, is insufficient to induce retrograde PHP expression.

**Chimeric GluRIIB subunits swapped with the GluRIIA C-tail recruit pCaMKII and suppress retrograde PHP signaling**

Although the loss of pCaMKII at postsynaptic compartments is insufficient to induce PHP expression, postsynaptic overexpression of constitutively active *CaMKII* in *GluRIIA* mutants appears to suppress the retrograde signaling required for PHP expression[28,30] (Fig. 1). Therefore, we sought to determine if recruitment of active pCaMKII at *GluRIIA* mutant NMJs was sufficient to occlude the signaling necessary for PHP expression.

To address this question, we generated chimeric *GluRIIB* receptor subunits in which the entire *GluRIIB* C-tail was replaced with the *GluRIIA* C-tail. This chimeric GluRIIB receptor subunit will be referred to as

GluRIIB^IIAtail (Fig. 7A). In *GluRIIA* null mutants, the entire postsynaptic receptive field is composed of GluRB receptors, mEPSPs are reduced, and PHP is expressed[21,46]. To mimic this condition, we expressed either wild-type *GluRIIB* or chimeric *GluRIIB^IIAtail* receptor subunits in a genetic background in which both endogenous *GluRIIA*- and *GluRIIB*-receptor subunits are absent (*IIA/IIB⁻/⁻*), leaving only GluRB receptors (Fig. 7B). *GluRIIA* mutants were indeed phenocopied in this condition, with the absence of GluRIIA expression and similar levels of GluRIIB expression (Fig. 7B, C). To confirm these receptors encoded either the GluRIIB or GluRIIA tail, we also stained with anti-GluRIIA^tail or anti-GluRIIB, where the antigen targets the terminal 15 amino acids of the GluRIIB C-tail[46]. As expected, with wild-type *GluRIIB* expression, we observed loss of both the anti-GluRIIA and anti-GluRIIA^tail signals (Fig. 7B, C), as expected. However, this relationship was reversed when chimeric *GluRIIB^IIAtail* receptor subunits were expressed, with increased anti-GluRIIA^tail signal and loss of both anti-GluRIIA and -GluRIIB signals (Fig. 7B, C). Interestingly, while the pCaMKII signal was absent in both *GluRIIA* null mutants and with *GluRIIB* expression, as expected, the pCaMKII signal was present at wild-type levels when chimeric *GluRIIB^IIAtail* receptor subunits were expressed (Fig. 7B, C). These results demonstrate that the GluRIIA C-tail is sufficient to recruit active pCaMKII at postsynaptic compartments even when the native GluRIIA receptor subunit is absent.

Finally, we considered two possibilities for whether PHP could be induced at NMJs expressing wild-type or chimeric GluRB receptors. First, we speculated that pCaMKII may simply be a marker of the GluRIIA C-tail, and its activity may not be involved in endogenous PHP signaling. In this scenario, constitutively active *CaMKII* expression may

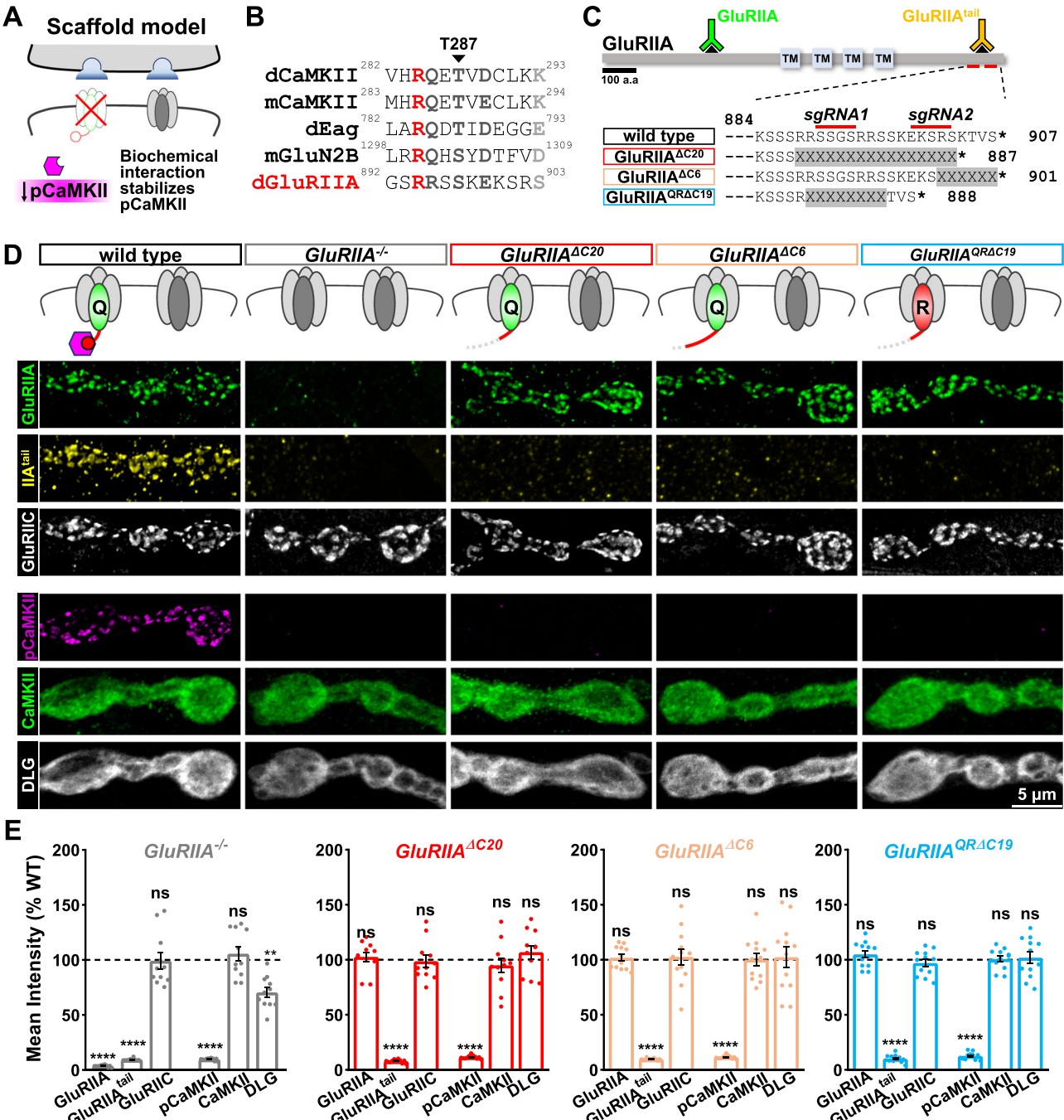

**Fig. 5 | Truncation of the GluRIIA C-tail abolishes active pCaMKII. A** Schematic illustrating the possibility that the GluRIIA C-tail stabilizes active pCaMKII at postsynaptic compartments. **B** Amino acid alignment of the mouse CaMKII autoinhibitory domain and homologous region in *Drosophila*, with the interaction sequences encoded in the C-tails of mouse GluN2B, Drosophila EAG, and the putative CaMKII interaction domain in the Drosophila GluRIIA C-tail. The black arrow indicates the Thr residue that is phosphorylated in active pCaMKII. The red Arg residue indicates the necessary R-X-X-S motif; dark gray indicates strongly conserved while light gray indicates weakly conserved residues. **C** Diagram of the GluRIIA C-tail amino acid sequence and three mutant truncation alleles of the C-tail induced by CRISPR mutagenesis. The regions of the two guide RNAs used to generate these alleles are shown, as well as the antigenic domains of two GluRIIA-specific antibodies. **D** Representative images of NMJ boutons immunostained with anti-GluRIIA, -GluRIIA^tail, -GluRIIC, -pCaMKII, -CaMKII, and -DLG antibodies in the indicated genotypes: *GluRIIA^ΔC20* (*w;GluRIIA^ΔC20*), *GluRIIA^ΔC9* (*w;GluRIIA^ΔC9*),

*GluRIIA^QRΔC19*, and (*w;GluRIIA^QRΔC19*). Experiments were repeated three times independently with similar results. Note that the loss of the terminal six amino acids of the GluRIIA C-tail is sufficient to completely lose pCaMKII signals. **E** Quantification of mean fluorescence intensities of the indicated antibody signal normalized to wild-type values (wild type: $n = 12$; GluRIIA^{-/-}, $n = 12$, $p < 0.0001$ for GluRIIA, $p < 0.0001$ for GluRIIA^tail, $p = 0.9912$ for GluRIIC, $p < 0.0001$ for pCaMKII, $p = 0.9958$ for CaMKII, $p = 0.0022$ for DLG; GluRIIA^ΔC20: $n = 12$, $p = 0.1744$ for GluRIIA, $p < 0.0001$ for GluRIIA^tail, $p = 0.9912$ for GluRIIC, $p < 0.0001$ for pCaMKII, $p = 0.9958$ for CaMKII, $p = 0.0022$ for DLG; GluRIIA^ΔC6: $n = 12$, $p = 0.6019$ for GluRIIA, $p < 0.0001$ for GluRIIA^tail, $p = 0.9912$ for GluRIIC, $p < 0.0001$ for pCaMKII, $p = 0.9958$ for CaMKII, $p = 0.0022$ for DLG; GluRIIA^QRΔC19: $n = 12$, $p = 0.9967$ for GluRIIA, $p < 0.0001$ for GluRIIA^tail, $p = 0.9912$ for GluRIIC, $p < 0.0001$ for pCaMKII, $p = 0.9958$ for CaMKII, $p = 0.0022$ for DLG). Error bars indicate ±SEM. *$p < 0.05$, ****$p < 0.0001$; ns, not significant. $n$ values indicate biologically independent NMJs. Source data are provided as a Source Data file.

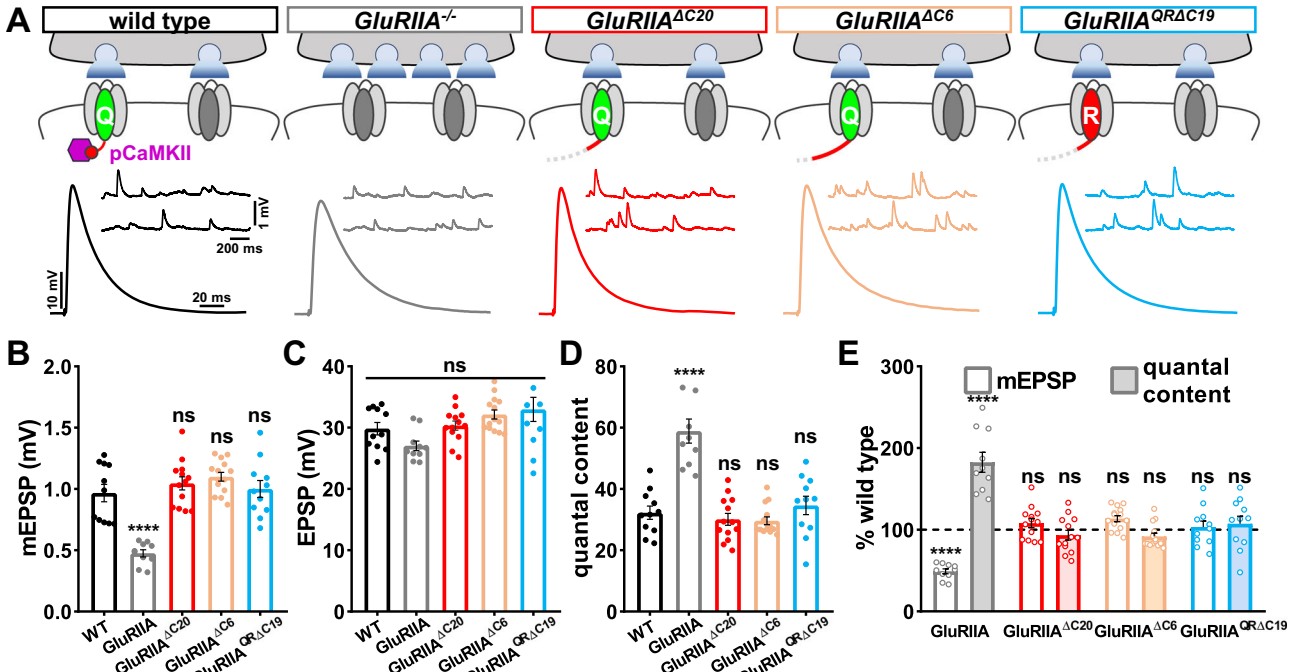

**Fig. 6 | Loss of active pCaMKII does not trigger retrograde homeostatic signaling. A** Schematics and representative traces show that loss of the GluRIIA C-tail does not alter spontaneous neurotransmission or presynaptic function.
**B–D** Quantification of average mEPSP amplitude (**B**), EPSP amplitude (**C**), and quantal content (**D**) values in the indicated genotypes shown in (**A**).
**E** Quantification of mEPSP and quantal content values of the indicated genotypes normalized to wild-type values (wild type: $n = 11$; GluRIIA$^{-/-}$, $n = 10$, $p < 0.0001$ for mEPSP, $p = 0.276$ for EPSP, $p < 0.0001$ for QC; GluRIIA$^{\Delta C20}$: $n = 13$, $p = 0.682$ for

mEPSP, $p = 0.992$ for EPSP, $p = 0.921$ for QC; GluRIIA$^{\Delta C9}$: $n = 14$, $p = 0.230$ for mEPSP, $p = 0.371$ for EPSP, $p = 0.852$ for QC; GluRIIA$^{QR\Delta C19}$: $n = 13$, $p = 0.980$ for mEPSP, $p = 0.176$ for EPSP, $p = 0.905$ for QC). Repeated measures one-way ANOVA with Dunnett's multiple comparisons test with a significance value of 0.05. $p$ Value adjusted for multiple comparisons. Error bars indicate ±SEM. *$p < 0.05$; ****$p < 0.0001$; ns, not significant. $n$ values indicate biologically independent NMJs. Absolute values for normalized data are summarized in Table S1. Source data are provided as a Source Data file. Source data are provided as a Source Data file.

block retrograde PHP signaling through a gain-of-function artifact, perhaps by non-specific phosphorylation of postsynaptic machinery that ends up perturbing homeostatic signaling. In contrast, we considered the possibility that a key event for PHP induction was the physical loss of the GluRIIA C-tail. This would lead to loss of pCaMKII and perhaps release a constitutive suppression of retrograde PHP signaling normally imposed by active pCaMKII.

To distinguish between these possibilities, we recorded from control *GluRIIA* mutants (*GluRIIA* null mutants and *GluRIIB* expression) or *GluRIIA* mutants composed of chimeric GluRB receptors. As expected, mEPSP amplitudes were reduced by over 50% in all three genotypes compared to the wild type (Fig. 8A–C). Also, as expected, EPSP amplitudes remained similar to wild type in *GluRIIA* mutants and *GluRIIB* expression due to enhanced presynaptic neurotransmitter release (quantal content), demonstrating robust PHP expression (Fig. 8A–E). However, no change in presynaptic neurotransmitter release was observed with chimeric *GluRIIB*$^{IIAtail}$ expression, leading to diminished EPSP amplitude and indicating a failure to express retrograde PHP signaling (Fig. 8A–E). Thus, the GluRIIA C-tail is sufficient to both activate pCaMKII and suppress retrograde PHP signaling at *GluRIIA* mutant NMJs. Importantly, the chimeric *GluRIIB*$^{IIAtail}$ condition is electrophysiologically identical to *GluRIIA* mutants, including the same reduction in mEPSP amplitude and Ca$^{2+}$ influx. This suggests that pCaMKII is intimately associated with the GluRIIA C-tail at postsynaptic compartments, where it exerts a constitutive suppression of retrograde PHP signaling. Loss of pCaMKII is therefore a key event necessary to disinhibit PHP signaling (schematized in Fig. 8F).

## Discussion

PHP was first described in 1997, where the genetic loss of the *GluRIIA* subunit reduced mEPSP amplitude but, surprisingly, synaptic

strength was unchanged from wild type[21]. It was immediately hypothesized that reductions in postsynaptic Ca$^{2+}$ levels, due to loss of *GluRIIA*, were the key inductive signal to initiate retrograde PHP signaling. Further studies of CaMKII at the *Drosophila* NMJ appeared to support this model[24,28–30], and speculation in favor of this prominent hypothesis has continued[5,18,41] despite a lack of direct evidence to support it. Here, we have interrogated this model and concluded that reduced Ca$^{2+}$ in postsynaptic compartments is not sufficient to induce PHP signaling, nor does synaptic Ca$^{2+}$ signaling have any apparent impact on the autoactivation state of CaMKII. Rather, our data support an alternative model in which CaMKII activation is entirely dependent on a small domain encoded in the C-tail of the GluRIIA receptor subunit, which in turn exerts a constitutive suppression of retrograde homeostatic signaling. Thus, a key event in enabling PHP is the recognition of the physical loss of the GluRIIA C-tail at postsynaptic compartments.

CaMKII is a central regulator of Hebbian plasticity at postsynaptic compartments in the mammalian brain. Three major roles for CaMKII have been described: Ca$^{2+}$ sensing during plasticity and learning, structural plasticity, and scaffolding. Dynamic changes in postsynaptic Ca$^{2+}$ are transformed into graded, longer-term responses through the activation of CaMKII[31]. Importantly, CaMKII activity is sensitive to the pattern of Ca$^{2+}$ changes in addition to the absolute amount, where such differences are thought to account for differential induction of long-term potentiation or depression[47]. Another layer of regulation is through an association with the NMDA receptor C-tail, where conformational changes induced by glutamate binding are critical to recruiting CaMKII to the postsynaptic compartment[48]. This preserves an active CaMKII state even after Calmodulin dissociation, facilitating autophosphorylation and "autonomous" activity[23]. In addition to these roles for CaMKII in

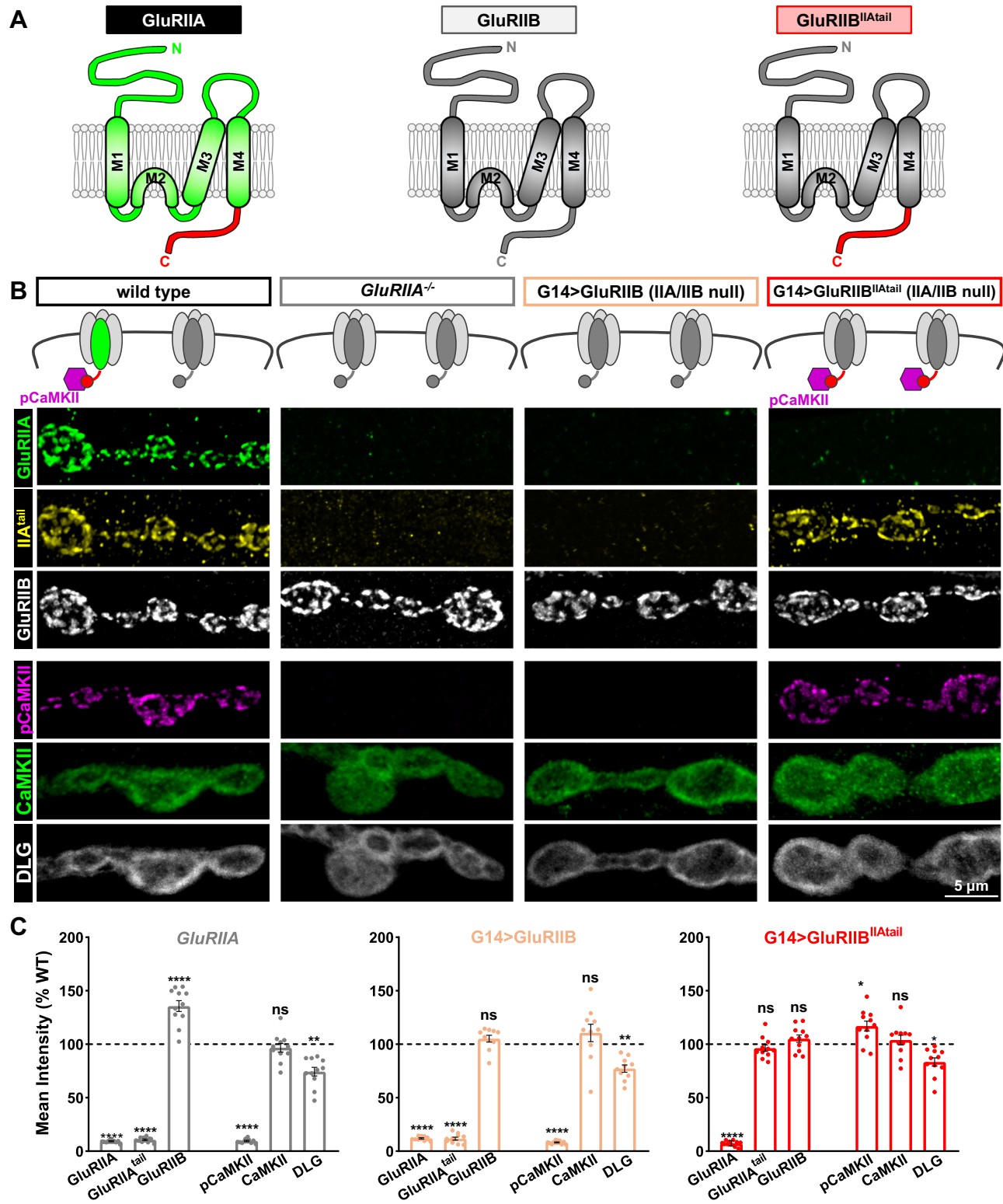

Hebbian plasticity and learning, CaMKII is important for structural plasticity at dendritic spines. Here, CaMKII interacts with the dendritic cytoskeleton to drive spine enlargement[49,50]. Finally, CaMKII serves as a scaffold to promote the assembly of signaling machinery in dendrites[51]. In contrast to the overwhelming evidence for CaMKII having crucial functions in Hebbian functional and structural plasticity at synapses, it is less clear to what extent CaMKII operates in either homeostatic synaptic plasticity or retrograde signaling at mammalian synapses.

Several lines of evidence suggest CaMKII is uniquely regulated at the *Drosophila* NMJ in the context of retrograde homeostatic signaling. First, while $Ca^{2+}$ levels are a major control mechanism to mobilize CaMKII to postsynaptic densities[23], levels of CaMKII and pCaMKII appear to be completely insensitive to $Ca^{2+}$ influx at the fly NMJ. This is illustrated most directly by the finding that pCaMKII levels are unchanged in *GluRIIA^{Q615R}* (Fig. 2) despite a major reduction in postsynaptic $Ca^{2+}$, and do not change in the absence of any synaptic glutamate release or $Ca^{2+}$ activity (Fig. 3). Second, while interactions with the

**Fig. 7 | Chimeric GluRIIB subunits containing the GluRIIA C-tail are able to recruit pCaMKII. A** Schematic illustrating the intracellular C-tail domains of GluRIIA, GluRIIB, and the chimeric GluRIIB subunit substituted with the GluRIIA C-tail (GluRIIB^IIAtail). Experiments were repeated three times independently with similar results. **B** Schematic and representative images of boutons stained with anti-GluRIIA, -GluRIIA^tail, -GluRIIB, -pCaMKII, -CaMKII, and -DLG antibodies at NMJs of wild type and those containing only GluRB receptors in the indicated genotypes: G14 > GluRIIB (IIA/IIB^−/−) (*w;G14-GAL4,GluRIIA^SP22/Df(2L)cl^h4;UAS-GluRIIB/*+), G14 > GluRIIB^IIAtail (IIA/IIB^−/−) (*w;G14-GAL4,GluRIIA^SP22/Df(2L)cl^h4;UAS-GluRIIB^IIAtail/*+). Experiments were repeated three times independently with similar results. Note that GluRB receptors containing the GluRIIA C-tail recruit pCaMKII at levels unchanged from the wild type. **C** Quantification of mean fluorescence intensity of the indicated

antibodies in the indicated genotypes normalized to wild-type values (wild type: $n = 13$; GluRIIA^−/−, $n = 11$, p < 0.0001 for GluRIIA, $p < 0.0001$ for GluRIIA^tail, $p < 0.0001$ for GluRIIB, $p < 0.0001$ for pCaMKII, $p = 0.9428$ for CaMKII, $p = 0.0012$ for DLG; G14 > GluRIIB: $n = 10$, p < 0.0001 for GluRIIA, $p < 0.0001$ for GluRIIA^tail, $p = 0.6436$ for GluRIIB, $p < 0.0001$ for pCaMKII, $p = 0.3599$ for CaMKII, $p = 0.0054$ for DLG; G14 > GluRIIB^IIAtail: $n = 11$, p < 0.0001 for GluRIIA, $p = 0.5391$ for GluRIIA^tail, $p = 0.6305$ for GluRIIB, $p = 0.0303$ for pCaMKII, $p = 0.9109$ for CaMKII, $p = 0.0468$ for DLG). Repeated measures one-way ANOVA with Dunnett's multiple comparisons test with a significance value of 0.05. $p$ Value adjusted for multiple comparisons. Error bars indicate ±SEM. Asterisks indicate statistical significance using One-way ANOVA: *$p < 0.05$, **$p < 0.01$, ****$p < 0.0001$; ns, not significant. $n$ values indicate biologically independent NMJs. Source data are provided as a Source Data file.

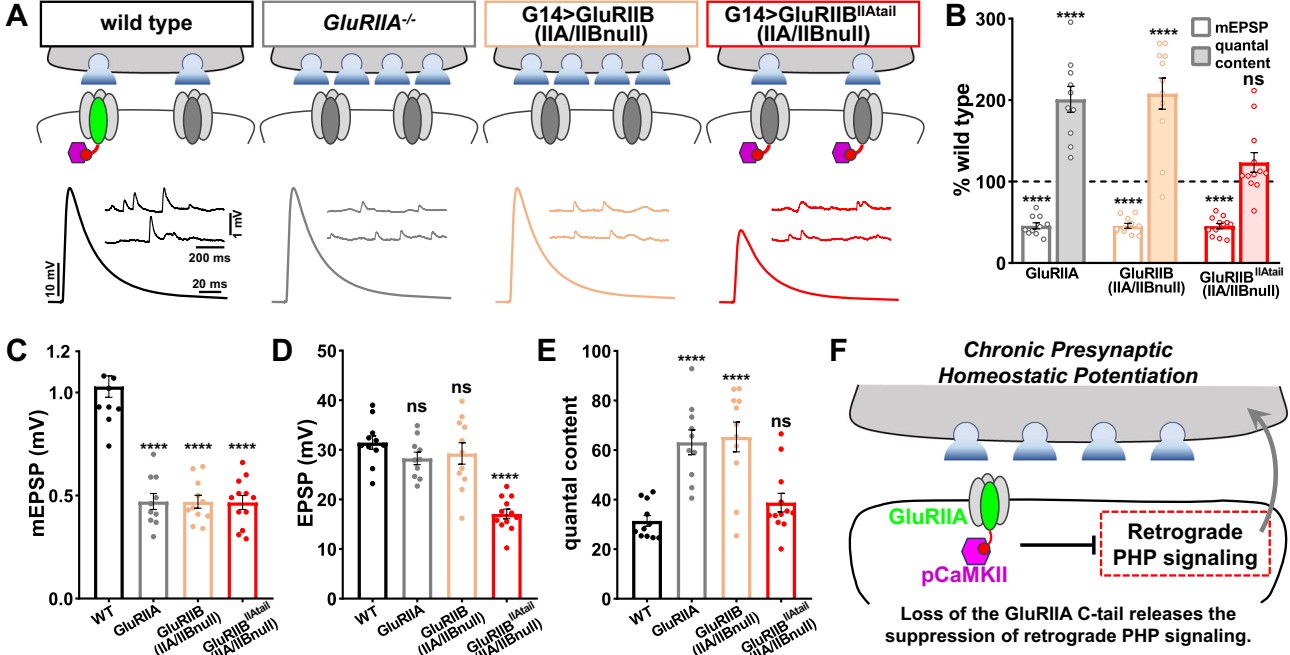

**Fig. 8 | Retrograde PHP signaling is occluded when pCaMKII is recruited to NMJs lacking GluRA receptors. A** Schematic and representative traces illustrating that retrograde PHP signaling is occluded at NMJs containing chimeric GluRB receptors, while PHP is robustly expressed at NMJs containing wild-type GluRB receptors. Note that while mEPSP amplitudes are similarly reduced at NMJs containing only GluRB receptors, the presynaptic release does not increase when pCaMKII is recruited to NMJs by chimeric GluRB receptors. **B** Quantification of mEPSP and quantal content values in the indicated genotypes normalized to wild-type values (wild type: $n = 12$; GluRIIA^−/−, $n = 10$, $p < 0.0001$ for mEPSP, $p < 0.0001$ for QC; G14 > GluRIIB: $n = 11$, $p < 0.0001$ for mEPSP, $p < 0.0001$ for QC; G14 > GluRIIB^IIAtail: $n = 12$, $p < 0.0001$ for mEPSP, $p = 0.4908$ for QC). **C–E** Quantification of average mEPSP amplitude (**C**), EPSP amplitude (**D**), and quantal content values (**E**)

in the indicated genotypes (wild type: $n = 12$; GluRIIA^−/−, $n = 10$, $p < 0.0001$ for mEPSP, $p = 0.331$ for EPSP, $p < 0.0001$ for QC; G14 > GluRIIB: $n = 11$, $p < 0.0001$ for mEPSP, $p = 0.596$ for EPSP, $p < 0.0001$ for QC; G14 > GluRIIB^IIAtail: $n = 12$, $p < 0.0001$ for mEPSP, $p < 0.0001$ for EPSP, $p = 0.490$ for QC). Repeated measures one-way ANOVA with Dunnett's multiple comparisons test with a significance value of 0.05. $p$ Value adjusted for multiple comparisons. **F** Schematic summarizing that active pCaMKII is lost from postsynaptic compartments due to the physical absence of the GluRIIA C-tail. This, in turn, releases the inhibition of retrograde PHP signaling. Error bars indicate ±SEM. ****$p < 0.0001$; ns, not significant. $n$ values indicate biologically independent NMJs. Absolute values for normalized data are summarized in Table S1. Source data are provided as a Source Data file.

NMDAR C-tail promote CaMKII activity along with Ca$^{2+}$ in dendrites[48], pCaMKII appears to absolutely require the GluRIIA C-tail, which is encoded by a kainate-type GluR with no sequence homology to NMDARs[32,34]. This implies an "all or nothing" binary switch between CaMKII activation states which is entirely dependent on the presence of the GluRIIA C-tail. In contrast, total CaMKII levels do not significantly change at postsynaptic NMJ compartments regardless of the state of GluRIIA or Ca$^{2+}$. These properties parallel associations between CaMKII and other synaptic components in regulating the transition of sustained CaMKII activity independently of Ca$^{2+}$/Calmodulin binding in *Drosophila*[44,52]. It is interesting to note that recent studies have demonstrated non-ionotropic signaling through GluRs[53], and CaMKII regulation by GluRIIA may be a new example of this mode of signaling. One intriguing possibility is that the interaction between CaMKII and

the GluRIIA C-tail regulates liquid-liquid phase separation with other postsynaptic components, as was recently shown in mammals[54]. Therefore, homeostatic plasticity at the NMJ utilizes a novel and unique mechanism to regulate CaMKII activity, distinct from the well-known paradigm illustrated by Hebbian plasticity in the mammalian brain.

Rather than functioning as a classical Ca$^{2+}$ sensor, we propose that CaMKII instead works as a "GluRIIA sensor" to constitutively inhibit retrograde PHP signaling at the fly NMJ. In this view, CaMKII does not operate as a conventional Ca$^{2+}$ sensor to monitor ionic activity at postsynaptic NMJ compartments but instead utilizes a docking or scaffolding function to recognize the physical presence of GluRA receptors. When the GluRIIA subunit is present, PHP is inhibited, and, in this overly simplified model, genetic loss of *GluRIIA* releases this inhibition. Clearly, there is more signal transduction necessary to

activate the retrograde signaling system necessary to express chronic PHP, with translational regulation appearing to play a key role[28,29]. However, one necessary step is the disinhibition of retrograde signaling through the loss of pCaMKII. The terminal portion of the GluRIIA C-tail may serve as a CaMKII docking site to not only stabilize active pCaMKII but also organize its signaling functions, paralleling CaMKII functions in dendritic spines[51]. Short peptide domains that interact with CaMKII typically behave as pseudosubstrates by interacting with the catalytic site to dislodge the Thr regulatory site and promote autonomous autophosphorylation[31]. The quintessential example of this type of peptide, the GluN2B C-terminal domain, contains an optimal CaMKII consensus sequence[47]. A similar CaMKII docking function has also been observed in the *Drosophila* Eag potassium channel[43]. A short stretch of the GluRIIA C-tail also encodes a conserved sequence with homology to these peptides, and CRISPR-mediated deletion of this sequence abolishes pCaMKII. Thus, this C-tail domain in the GluRIIA subunit is an attractive direct target for CaMKII interaction and regulation. However, it is possible that CaMKII may interact with the GluRIIA C-tail indirectly, perhaps through other postsynaptic components tightly associated with GluRs, such as the auxiliary GluR subunit Neto or postsynaptic scaffold DLG. Indeed, in rodents, CaMKII phase separates with PSD95 and the auxiliary receptor subunit Stargazin[54].

The role we have described here for CaMKII is likely to be specific to chronic PHP inductive signaling. For the rapid, pharmacological induction of PHP, a distinct process is likely involved. There is substantial evidence to indicate that disparate postsynaptic signaling systems operate to enable chronic PHP expression (due to genetic loss of *GluRIIA*) vs rapid PHP (following pharmacological blockade of GluRs[4,29,30]). For example, some genes are necessary only for chronic PHP, while they are dispensable for rapid PHP[4]. Furthermore, while translational regulation is necessary for chronic PHP[55–57], rapid PHP does not require new protein synthesis[29,41,58,59]. An important component of the postsynaptic signaling system that regulates both chronic and rapid PHP is mono-ubiquitination by the ubiquitin ligase Cul3 and its adapter Insomniac[60]. While we hypothesize that chronic PHP induction requires loss of the GluRIIA C-tail and pCaMKII, clearly, rapid PHP would necessitate a distinct mechanism since the GluRIIA tail still remains present. However, one important commonality between rapid and chronic PHP is that reduced postsynaptic $Ca^{2+}$ does not seem to be involved in either process[29]. Elucidating the induction mechanism of rapid PHP, and determining to what extent CaMKII is involved, will be an exciting area of future research.

Our study not only reveals a novel interaction between postsynaptic GluRs and CaMKII regulation at the NMJ, but highlights that PHP, and perhaps other types of homeostatic plasticity, functions independently of $Ca^{2+}$ signaling. At dendrites of glutamatergic synapses in the brain, Hebbian and homeostatic plasticity mechanisms work in conjunction to calibrate synaptic strength and efficacy to enable the flexibility necessary for learning and memory while preventing runaway excitation[61]. In this context, it would seem advantageous to use $Ca^{2+}$ as a common signal to integrate the signal transduction and crosstalk between various forms of plasticity. However, the NMJ may not require such integration, since potent homeostatic signaling systems stabilize this synapse to coordinate locomotion through muscle contraction, being essential for behavior and life, while Hebbian plasticity at this NMJ is more limited but has been observed[62]. An additional contrast is that while Hebbian plasticity and homeostatic receptor scaling are bi-directionally expressed at dendritic spines[63], PHP appears to be uni-directional[4]. Although the loss of NMJ receptor functionality can clearly lead to motor dysfunction, increased depolarization of the muscle is tolerated, given the safety factor characteristic of all NMJs[64]. Thus, the unique characteristics of the NMJ may enable the discovery of synaptic plasticity mechanisms that may not be as readily apparent at central synapses.

## Methods

### Fly strains

Experimental flies were raised at 25 °C on standard molasses food. The $w^{1118}$ strain was used as the wild-type control unless otherwise noted, as this is the genetic background for which all genotypes are bred. The following fly strains were generated in this study: *UAS-PV*, *UAS-GluRIIA*, *UAS-GluRIIB*, *UAS-GluRIIB^{IIAtail}*, *GluRIIA^{Q615R}*, *GluRIIA^{ΔC20}*, *GluRIIA^{ΔC6}*, *GluRIIA^{QRΔC19}*, and *MHC-CD8-GCaMP8f-Sh* (SynapGCaMP8f). All details about fly stocks and other reagents used in this study can be found in the Key Resources Table (Table S2).

### Molecular biology

To generate the *UAS-PV* and *UAS-GluRIIB* transgenes, we obtained the cDNAs of PV from Addgene (#17301) and GluRIIB from the Drosophila Genomics Resource Center (DGRC #1374682). We inserted the *PV* and *GluRIIB* cDNA sequences into the pACU2 vector[37] (#31223; Addgene). To generate SynapGCaMP8f (*MHC-CD8-GCaMP8f-Sh*), we obtained the SynapGCaMP6f transgenic construct[24] and replaced the sequence encoding GCaMP6f with the GCaMP8f[40] sequence (#162379; Addgene) using Gibson assembly as described[39]. Transgenic stocks were generated by Bestgene, Inc (Chino Hills, CA 91709, USA) and inserted into $w^{1118}$ (#5905, BDSC) fly strains by P-element-mediated random insertion. To generate the *UAS-GluRIIB^{IIAtail}* transgenes, the 5′ fragments of *GluRIIB* (1–2511 bp) and the 3′ fragment of *GluRIIA* (2510–2724 bp) were cloned from the cDNAs of *GluRIIB* and *GluRIIA*[65]. The *GluRIIB^{IIAtail}* fragment was then generated by overlap extension PCR from the *GluRIIB* 5′ fragment and the *GluRIIA* 3′ fragment. The *GluRIIB^{IIAtail}* fragment was then inserted into the pUAST vector[66]. The transgenic stock of *UAS-GluRIIB^{IIAtail}* was generated by Eppendorf InjectMan (Hamburg, Germany) and inserted into the $w^{1118}$ strain.

### CRISPR/Cas9 mutagenesis

To generate the $Ca^{2+}$ impermeable *GluRIIA^{Q615R}* allele, a sequence containing 1 kb homology arms of the *GluRIIA* genomic region with the Q615R point mutation was inserted into pHD-DsRed vector (#51434; Addgene) as the CRISPR donor. Two single guide RNAs (gRNA1: gaacaactcgacttggctga, gRNA2: ggtgggctccatcatgcaac) were inserted together into the pAC-U63-tgRNA (#112811; Addgene) vector with intervening tRNA(F + E) sequences for expressing multiple gRNAs[67]. The donor construct and the gRNA construct were then co-injected into a nos-Cas9 (#78782; BDSC) fly strain by Well Genetics (Taipei City, Taiwan (R.O.C.)) to generate the *GluRIIA^{Q615R}* mutant by homology-directed repair. Successful CRISPR fly lines were selected by P3 > DsRed expression in eyes and confirmed by PCR. DsRed with flanking PBac sequence was then removed by PBac-mediated excision suing the Tub > PBac fly strain (#8283, BDSC).

To generate endogenous *GluRIIA* tail truncations, two independent single guide RNAs (sgRNAs; gRNA1: tctggaaccggatgatcgcc, gRNA2: ggaaaagtcccgcagcaaga) were inserted together into the pAC-U63-tgRNA vector. The construct was then injected and inserted into the attP2 (#8622, BDSC) fly strain by phiC31 integration. Fly strains carrying this transgene were crossed to nos-Cas9 (#78782; BDSC) to generate putative truncation alleles, and *GluRIIA^{ΔC20}* and *GluRIIA^{ΔC6}* alleles were confirmed by PCR. *GluRIIA^{QRΔC19}* was generated using a similar approach but crossed to the *GluRIIA^{Q615R}* strain.

### Electrophysiology

Third-instar larvae were dissected in ice-cold modified HL3 saline as described[68,69]. Briefly, modified HL3 saline contained (in mM): 70 NaCl, 5 KCl, 10 $MgCl_2$, 10 $NaHCO_3$, 115 sucrose, 5 trehalose, and 5 HEPES at pH 7.2. Guts, trachea, and the central nervous system were removed from the larval body wall. The preparation was perfused three times with fresh HL3 saline. For mEPSP and EPSP recordings, sharp electrode (electrode resistance between 10-35 MΩ) recordings were performed on body wall muscle 6 of segment A2 and A3 in HL3 saline with 0.4 mM

CaCl$_2$ added. Recordings were conducted using an Olympus BX61 WI microscope with a 40x/0.80 water-dipping objective and acquired using an Axoclamp 900 A amplifier, Digidata 1440 A acquisition system and pClamp 10.5 software (Molecular Devices). To stimulate evoked EPSPs in muscles, 20 electrical stimulations at 0.5 Hz with 0.5 ms duration were delivered to motor neurons using an ISO-Flex stimulus isolator (A.M.P.I.) with stimulus intensities set to avoid multiple EPSPs. Electrophysiological signals were digitized at 10 kHz and filtered at 1 kHz. Recordings were rejected with input resistances lower than 5 Ωohm or resting potentials more depolarized than −60 mV. Data were analyzed using Clampfit (Molecular Devices), MiniAnalysis (Synaptosoft), or Excel (Microsoft). Average mEPSP, EPSP, and quantal content values were calculated for each genotype.

## Immunocytochemistry

Third-instar larvae were dissected in modified HL3 saline and stained either with or without 0.03% Triton in PBS as described[68]. The following primary antibodies were used: mouse anti-GluRIIA (8B4D2; 1:50; Developmental Studies Hybridoma Bank (DSHB)); rabbit anti-GluRIIIB[70] (1:1000); rabbit anti-GluRIIC[71] (1:2000); guinea pig anti-GluRIID[70] (1:1000); rabbit anti-parvalbumin (Pa1-933; 1:1000; Thermo Fisher); mouse anti-DLG (4F3; 1:100; DSHB). The following primary antibodies were generated in this study, where the following peptides were injected into animals by Cocalico Biologicals (Stevens, PA, U.S.A): affinity purified rabbit anti-pCaMKII using the peptide C-VHRQET(p)VDCLKK (1:2000); guinea pig anti-CaMKII using the peptide C-VHRQET(p)VDCLKK (1:1000); guinea pig anti-GluRIIA$^{tail}$ using the peptide C-SGSRRSSKEKSRSKTVS (1:2000). Alexa Fluor-647 conjugated goat anti-HRP (1:200; Jackson ImmunoResearch) and Donkey anti-mouse, -guinea pig, and -rabbit conjugated Alexa Fluor 488, Cy3, and DyLight 405 secondary antibodies (Jackson ImmunoResearch) were used at 1:400. For the BAPTA-AM treatment, dissected larvae were incubated in HL-3 saline with 0.1 or 0.3 mM BAPTA-AM (#120503; Abcam) for 30 min before fixation and then stained as described above. For the control conditions, dissected larvae were incubated in HL-3 saline in 0 of 1.8 mM Ca$^{2+}$.

## Immunoblots

Third-instar larval muscle extracts (12 dissected body walls of each genotype) were prepared and used for immunoblotting, as previously described[36]. Larval body wall muscles were homogenized in ice-cold RIPA buffer (Cell Signaling Technology) mixed with an EDTA-free protease inhibitor cocktail (Thermo Scientific) and run on 4-12% Bis−Tris Plus gels. After blotting onto PVDF membrane (Novex) and incubated with 5% nonfat milk in TBST (Thermo Scientific, with 5% Tween 20) for 60 min, the membrane was washed once with TBST and incubated with primary antibodies at 4°C overnight. The following antibodies were used: mouse anti-DLG (4F9, 1:1000, Developmental Studies Hybridoma Bank, USA), guinea pig anti-CaMKII (1:1000, this study), mouse anti-β-tubulin (E7, 1:200; Developmental Studies Hybridoma Bank, USA). Membranes were washed three times and incubated with a 1:5000 dilution of horseradish peroxidase-conjugated anti-mouse or anti-guinea pig secondary antibodies (Jackson ImmunoResearch) for 1 h. Blots were washed with TBST and visualized with the ECL Prime Western Detection Reagent (Amersham) and exposed to G:BOX Chemi XX6 (Syngene). Bands intensities were determined with ImageJ (NIH) using the gel analysis plug-in.

## Quantitative PCR

Quantitative PCR (qPCR) was performed as described[72] using the Luna Universal One-Step RT-qPCR Kit (NEB, E3005S) according to the manufacturer's instructions. RNA was isolated and prepared from body wall tissue using a standard phenol/chloroform extraction and DNAse1 treatment. Total, 60 ng of total RNA was used as a template in each reaction. Three technical replicates were performed for each sample, and the 2$^{-\Delta\Delta Ct}$ method was used for qPCR data analysis. The previously validated primers used for assaying each target are as follows (fwd/rev, 5′-3′): Tub84D (control): CTACAACTCCATCCTAACCACG; CAGGTTAGTGTAAGTGGGTCG; CaMKII: AAAGGAGCCCTATGGGAAATCG; CCCAAAAGGGTGGATAAC.

## Confocal imaging and analysis

Samples were imaged using a Nikon A1R Resonant Scanning Confocal microscope equipped with NIS Elements software and a 100× APO 1.4 NA or 60× 1.4 oil immersion objective using separate channels with four laser lines (405, 488, 561, and 637 nm) as described[68]. All genotypes compared were immunostained in the same tube, mounted, and imaged using the same procedure with identical reagents. Z-stacks were acquired with a step size of 200 nm and pixel size of 0.06 nm using the same setting across all genotypes compared. Maximum intensity projections were applied for quantitative image analysis of Type Ib motor neuron boutons using Nikon Element software. Puncta of anti-GluRIIA, -GluRIIB, -GluRIIC, -GluRIID, -DLG, -pCaMKII, -CaMKII, and GluRIIA$^{tail}$ were detected using thresholding of fluorescence intensity and object size to generate binary objects in the general analysis tool kit in Nikon Elements software as described[73]. Identical parameters for image analysis were applied across all samples compared.

## Ca$^{2+}$ imaging and analysis

Third-instar larvae were dissected in ice-cold modified HL3 saline. Larval preparations were imaged using an A1R Resonant Scanning Confocal microscope equipped with NIS Elements software and a 60x APO 1.0NA water immersion objective as detailed[30]. Imaging was performed in modified HL3 saline with 1.5 mM Ca$^{2+}$ added. NMJs on muscle 6/7 were imaged with band scanning at a resonant frequency of 100 fps (512 × 86 pixels). Spontaneous Ca$^{2+}$ events were recorded at 4–8 individual NMJs during 120 s imaging sessions from at least two different larvae. Horizontal drifting was corrected using ImageJ plugins[74], and imaging data with severe muscle movements were rejected as described[75]. Three ROIs were manually selected using the outer edge of terminal Ib boutons observed by baseline GCaMP signals with ImageJ[76,77]. Ib and Is boutons were defined by baseline GCaMP6f fluorescence levels, which are 2–3 fold higher at Ib NMJs compared to their Is counterparts at a particular muscle. Fluorescence intensities were measured as the mean intensity of all pixels in each individual ROI. $\Delta F$ for a spontaneous event was calculated by subtracting the baseline GCaMP fluorescence level F from the peak intensity of the GCaMP signal during each spontaneous event at a particular bouton. Baseline GCaMP fluorescence was defined as average fluorescence in 2 s of each ROI without spontaneous events. $\Delta F/F$ was calculated by normalizing $\Delta F$ to baseline signal F. For each ROI under consideration, the spontaneous event $\Delta F/F$ value was averaged for all events in the 60 s time range to obtain the mean quantal size for each bouton. For imaging of firing patterns, SynapGCaMP8f signals were acquired in semi-intact preparations, including the brain and motor nerves at muscle 6 Ib boutons in the same conditions and settings described above for imaging of spontaneous signals. Images were acquired when spontaneous firing persisted after synchronized muscle contractions. Data analysis was performed with customized Jupyter Note codes[39].

## Statistical analysis

Data were compared using either a one-way ANOVA followed by Dunnett multiple comparison test or a Student's $t$-test (where specified) and analyzed using Graphpad Prism or Microsoft Excel software. $p$-Value and standard error of the mean (SEM) were reported. $p$-Values were adjusted by multiple testing corrections (Dunnett) when applicable.

## Reporting summary

Further information on research design is available in the Nature Portfolio Reporting Summary linked to this article.

## Data availability

The authors declare that the data supporting the findings of this study are available within the paper, the Source Data file, and the Supplementary Information and Supplementary Data files. Source data are provided in this paper.

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

## Acknowledgements

We thank Chun Han (Cornell University, USA), Ehud Isacoff (UC Berkeley, USA), and Leslie Griffith (Brandeis University, USA) for sharing valuable reagents. We acknowledge the Developmental Studies Hybridoma Bank (Iowa, USA) for antibodies used in this study and the Bloomington Drosophila Stock Center for fly stocks (NIH P40OD018537). We thank Beril Kiragasi for their technical contributions to this project at earlier stages, and Ying Wu, Mark Colt, and Karen Chang (University of Southern California, USA) for technical assistance with CaMKII immunoblots. S.P. was supported by an NRSA grant from NIGMS (GM130108). This work was supported by a grant from the National Institutes of Health (NS091546) to D.D.

## Author contributions

S.P. and D.D. conceived the project and planned the initial stages. S.P., Y.H., C.Q., C.C., P.G., S.N., and M.S. performed experiments and analyzed data. A.S. and S.J.S contributed chimeric *GluRIIB* transgenic flies. S.P., Y.H., and D.D. wrote the paper with feedback from the other authors.

## Competing interests

The authors declare no competing interests.
