## [Peer Review File · Nature Communications]

A Glutamate Receptor C-tail Recruits CaMKII to Suppress Retrograde Homeostatic SignalingReviewers' comments:

Reviewer #1 (Remarks to the Author):

A Glutamate receptor C-tail Recruits CaMKII to suppress Retrograde Homeostatic Signaling.

In the current study, Perry et al. cleverly interrogated the postsynaptic mechanisms hypothesized to initiate retrograde PHP signaling at the *Drosophila* NMJ. To examine the roles of GluRIIA, postsynaptic Ca²⁺ and CamKII activity in retrograde PHP induction the authors developed specific antibodies selective for *Drosophila* GluRIIA, phospho- and total CaMKII, as well as mutant alleles using CRISPR/Cas9. With these tools, the author demonstrate that CaMKII activity and PHP induction are insensitive to reductions in postsynaptic Ca²⁺ levels. Interestingly, the authors show that CaMKII activity is tightly linked to the C-terminal tail of the GluRIIA receptor. Using chimeric receptors, Perry et al show that the GluRIIA tail is sufficient to stabilize active pCaMKII and occlude retrograde homeostatic signaling. Their data support a model where retrograde PHP signaling is induced by postsynaptic mechanisms that sense the loss of the GluRIIA tail and not attenuated Ca signaling.

The data presented in this manuscript interrogate the long hypothesized, yet insufficiently tested, model of retrograde PHP signaling at the *Drosophila* NMJ. Perry et al. thoughtfully designed experiments to directly test this model and the roles for Ca, CaMKII and GluRIIA in retrograde PHP. Based on their data the authors conclude that reduced postsynaptic Ca is insufficient to alter CaMKII activity as well as to induce PHP. Rather, the data presented in this manuscript support a model where activation of CaMKII is dependent on the C-tail of GluRIIA, which in turn, exerts a constitutive suppression of retrograde homeostatic signaling. These data highlight a unique Ca-independent regulatory mechanism for CaMKII in homeostatic plasticity. The experiments described in this manuscript are thoughtfully designed, well controlled and compelling. The figures are also very clear and easy to follow.

Minor points:

1. In Fig. 1 the authors show overexpression of CaMKII-T287D (constitutively active) occludes PHP in GluRIIA mutants, implicating reduced postsynaptic CaMKII activity in induction of retrograde PHP signaling. Considering the surprising and compelling mechanism proposed by the authors in Fig. 7, what happens to PHP when a kinase-dead/inactive CaMKII is overexpressed in the GluRIIA mutants?
2. In Fig. S1, the authors show that when they overexpress the CaMKII inhibitory peptide (Ntide) pCaMKII decreases and GluRIIA levels increase (S1A-B, third panel). The increase in GluRIIA is significant and yet not discussed. Why does GluRIIA increase under these conditions but not in the CaMKII-Ala mutant? Discussion of these data would be useful.
3. Fig. 2, the authors show that pCaMKII levels are insensitive to reductions in postsynaptic Ca levels in GluRIIA-Q615R mutants. Since retrograde PHP can be induced by inhibition of GluRA receptors, what happens to postsynaptic Ca levels, pCaMKII, total CaMKII levels under these conditions? Does inhibition of GluRA receptors in the GluRII-Q615R reduce Ca levels below that observed for the Q615R mutant alone? Could the observed insensitivity of pCaMKII to Ca levels be because sufficient Ca remains in the postsynaptic terminal?

Other points:

1. It would be helpful for the reader if the authors described the mutants used in Fig. S1, (eg. CaMKII-Ntide, CaMKII-ala, etc) in the figure legend and/or text with the name of the mutant and the associated phenotype (kinase dead etc) so a non-CaMKII expert could understand the data more easily (eg. why pCaMKII decreases in CaMKII-Ntide; CaMKII-ala = does what?).
2. The cartoons and graphs are a clever and clear way to show complex experiments. For readers that might not be able to distinguish the colors, it would be helpful to also label the bar graphs below each sample or have a legend next to them for extra clarity (eg. Fig 3).

Reviewer #2 (Remarks to the Author):

Presynaptic homeostatic plasticity is thought to be important for neuronal function. Here Perry et al report that the c-tail of GluRIIA is critical for recruiting activated CaMKII, which mediates PHP signaling. Some results of this manuscript, especially the part showing that c-tail of GluRIIA is important for docking CaMKII, is interesting, and the related experiments were well designed and solid. However, the conclusion regarding whether CaMKII activity is sensitive to reductions of postsynaptic calcium warrants further study. Based on their results, it is very likely that the conformational changes of c-tail in GluRIIA is important for recruiting activated CaMKII for PHP. If this is the case, however, this result is largely expected as it is exactly the way how postsynaptic CaMKII gets activated in mammals, which unfortunately reduces the significance of this manuscript for publication in Nature Communications.

1: The main findings are based on new antibodies developed by the authors. However, RNAi results show that pCaMKII or CaMKII staining only decreased ~50%, which makes me worry about whether these antibodies are specific. The authors should knock out CaMKII in their system and test the specificity of those antibodies. It is also more worrying as the authors saw that CaMKII intensity decreased when they overexpress CaMKII T287D (Fig, S1C, D).

2. CaMKII activation is a dynamic process, which is known to be triggered by neuronal activities. There are many factors that may affect CaMKII activation, such as the frequency of neuronal firing, calcium influx, and conformational changes of glutamate receptors. Indeed, it has been well known in mammals that calcium is necessary, but not sufficient for inducing postsynaptic CaMKII activation, as activity-dependent conformational changes in NMDARs are also required for this process. Therefore, the authors should examine the frequency of postsynaptic calcium transients to see whether it is affected by their genetic manipulations and fix the time point for measuring pCaMKII for each neuronal activation. Moreover, reducing ~50% calcium does not mean too much, as it is likely that the 50% calcium is still enough to trigger CaMKII activation. They may use calcium chelators such as BAPTA-AM to buffer all calcium influx and examine whether CaMKII can be still activated.

RESPONSE TO REVIEWS

RESPONSE TO REVIEWER 1

Reviewer 1 states: *In the current study, Perry et al. cleverly interrogated the postsynaptic mechanisms hypothesized to initiate retrograde PHP signaling at the Drosophila NMJ. To examine the roles of GluRIIA, postsynaptic Ca²⁺ and CamKII activity in retrograde PHP induction the authors developed specific antibodies selective for Drosophila GluRIIA, phospho- and total CaMKII, as well as mutant alleles using CRISPR/Cas9. With these tools, the author demonstrate that CaMKII activity and PHP induction are insensitive to reductions in postsynaptic Ca²⁺ levels. Interestingly, the authors show that CaMKII activity is tightly linked to the C-terminal tail of the GluRIIA receptor. Using chimeric receptors, Perry et al show that the GluRIIA tail is sufficient to stabilize active pCaMKII and occlude retrograde homeostatic signaling. Their data support a model where retrograde PHP signaling is induced by postsynaptic mechanisms that sense the loss of the GluRIIA tail and not attenuated Ca signaling.*

The data presented in this manuscript interrogate the long hypothesized, yet insufficiently tested, model of retrograde PHP signaling at the Drosophila NMJ. Perry et al. thoughtfully designed experiments to directly test this model and the roles for Ca, CaMKII and GluRIIA in retrograde PHP. Based on their data the authors conclude that reduced postsynaptic Ca is insufficient to alter CaMKII activity as well as to induce PHP. Rather, the data presented in this manuscript support a model where activation of CaMKII is dependent on the C-tail of GluRIIA, which in turn, exerts a constitutive suppression of retrograde homeostatic signaling. These data highlight a unique Ca-independent regulatory mechanism for CaMKII in homeostatic plasticity. The experiments described in this manuscript are thoughtfully designed, well controlled and compelling. The figures are also very clear and easy to follow.

We thank the reviewer for his/her positive assessment of our study.

Minor points:

1. *In Fig. 1 the authors show overexpression of CaMKII-T287D (constitutively active) occludes PHP in GluRIIA mutants, implicating reduced postsynaptic CaMKII activity in induction of retrograde PHP signaling. Considering the surprising and compelling mechanism proposed by the authors in Fig. 7, what happens to PHP when a kinase-dead/inactive CaMKII is overexpressed in the GluRIIA mutants?*

This is an astute question raised by the reviewer and one we have spent considerable time investigating. To restate the question raised by the reviewer - does inhibition of CaMKII activity alter PHP expression as observed in *GluRIIA* mutants? It was originally reported that postsynaptic overexpression of inhibitory peptides targeting CaMKII (UAS-Ala and UAS-N-tide) in an otherwise wild-type background led to a small but significant increase in EPSP amplitude (Haghighi et al, *Neuron* 2003). The authors of this study did not report postsynaptic overexpression of inactive CaMKII (UAS-CaMKII^{T287A}), nor did they express any of these transgenes in *GluRIIA* mutant backgrounds.

In early stages of this project initiated over four years ago, we attempted to repeat these experiments and obtained somewhat conflicting results. As we show in Supplemental Figure S1, postsynaptic overexpression of Ala or N-tide have the expected impacts on pCaMKII/CaMKII staining. However, while expression of UAS-Ntide did lead to a small but significant elevation in EPSP amplitude in our experiments (but no significant change in quantal content), expression of UAS-Ala had no impact, nor did UAS-CaMKII^{T287A} have any impact on transmission (**Reviewer Fig. 1 below**). We also observed normal PHP expression when these transgenes were expressed following GluRIIA knock-down by RNAi (Reviewer Fig. 1). The reason for our results not matching those published in the previous study are unclear. Because overexpression of peptides and CaMKII transgenes could lead to artifacts due to

Reviewer Figure 1: Inhibition of postsynaptic CaMKII activity does not impact presynaptic function or homeostatic plasticity. (A) Schematics and electrophysiological traces of the indicated genotypes. **(B)** Quantification of mEPSP and quantal content values normalized to baseline, confirming robust PHP expression in all genotypes. **(C)** Baseline EPSP, mEPSP, and quantal content values in the indicated genotypes.

disruptions in postsynaptic signaling (and/or possibly gain-of-function changes), we preferred not to include these types of experiments in our study (beyond the occlusion of PHP shown in Fig. 1 that serves to frame the original motivation of our project). However, if the reviewer prefers, we would be happy to include this data in the final manuscript as a supplemental figure and discuss the appropriate caveats in interpreting these results.

2. In Fig. S1, the authors show that when they overexpress the CaMKII inhibitory peptide (Ntide) pCaMKII decreases and GluRIIA levels increase (S1A-B, third panel). The increase in GluRIIA is significant and yet not discussed. Why does GluRIIA increase under these conditions but not in the CaMKII-Ala mutant? Discussion of these data would be useful.

This is another good point raised by the reviewer. It is not clear why CaMKII.Ntide overexpression leads to a small but significant increase in GluRIIA levels, a change that is not observed in Ala expression (Fig. S1A,B). It is possible that CaMKII.Ntide is a more potent inhibitor of CaMKII than Ala; during development of the postsynaptic SSR, significant inhibition of CaMKII signaling may lead to defects in the organization of the postsynaptic compartment, which may in turn alter the level of GluRIIA. Indeed, postsynaptic machinery, including the GluR scaffold DLG/PSD95, are phosphorylation targets of CaMKII (Kim and Sheng, *Nat Reviews Neuroscience* 2004; Koh et al., *Cell* 1999; Mauceri et al., *JBC* 2004; Steiner et al., *Neuron* 2008). We have added a brief discussion of this point in the **Fig. S1 legend** and thank the reviewer for these comments.

3. Fig. 2, the authors show that pCaMKII levels are insensitive to reductions in postsynaptic Ca levels in GluRIIA-Q615R mutants. Since retrograde PHP can be induced by inhibition of GluRA receptors, what happens to postsynaptic Ca levels, pCaMKII, total CaMKII levels under these conditions? Does inhibition of GluRA receptors in the GluRII-Q615R reduce Ca levels below that observed for the Q615R mutant alone? Could the observed insensitivity of pCaMKII to Ca levels be because sufficient Ca remains in the postsynaptic terminal?

These are interesting questions raised by the reviewer. Regarding the first set of questions, we assume that the “inhibition of GluRA receptors” referred to by the reviewer relates to pharmacological inhibition of GluRA receptors using the toxin philanthotoxin-433 (PhTx). Because PhTx appears to be a specific antagonist against GluRA receptors (Frank et al., *Neuron* 2006; Kiragasi et al., *Cell Reports* 2017) and since we demonstrate *GluRIIA^{Q615R}* mutant alleles do not pass Ca^{2+} and effectively phenocopy GluRIIA mutants (Fig. 2), no change in Ca^{2+} influx in *GluRIIA^{Q615R}* mutants after PhTx application would be predicted. Indeed, when we apply PhTx to *GluRIIA^{Q615R}* mutants, we observe no significant difference in Ca^{2+} influx compared to *GluRIIA^{Q615R}* mutants alone (Reviewer Fig. 2). We also observe that PHP is normally expressed (Reviewer Fig. 3).

The answer to the reviewer’s question about whether PhTx application to *GluRIIA^{Q615R}* mutants alters pCaMKII levels is more complicated. PhTx application to wild-type NMJs leads to remodeling of many components of the postsynaptic apparatus, including DLG (data not shown), and pCaMKII itself is reduced by ~40% (Reviewer Fig. 4). As we briefly outline in the discussion of the manuscript

(lines 446-460), the rapid induction and expression of PHP by PhTx is fundamentally different from chronic PHP induction in *GluRIIA* mutants. In fact, rapid PHP is robustly expressed when PhTx is applied to postsynaptic overexpression of *CaMKII^{T287D}* (preliminary data not shown), which in contrast blocks chronic PHP expression in *GluRIIA* mutants (Fig. 1). We are working on an ongoing project to understand how PhTx induces PHP, but the findings are too preliminary at this time to present a mature model or hypothesis; this will be the subject of a future study dedicated to the mechanism of pharmacological induction of PHP by PhTx and how changes in postsynaptic structure are involved. What we can say, however, is that PhTx application to *GluRIIA^{Q615R}* mutants has the same impact on pCaMKII as it does on wild type GluRA receptors (Reviewer Fig. 4).

In response to the final question raised by the reviewer about whether residual Ca^{2+} in the postsynaptic compartment could be sufficient to maintain levels of active pCaMKII, this was a major concern also shared by Reviewer 2. In response, we have provided an entirely new Figure in this revised manuscript (Fig. 3) where we have blocked all synaptic activity (using a novel Botulinum toxin transgene we have developed). Despite the block in all synaptic and Ca^{2+} activity induced by BoNT expression, pCaMKII levels remain unchanged compared to wild type (discussed in lines 220-238 and presented in Fig. 3).

Reviewer Figure 4: PhTx application to *GluRIIA^{Q615R}* NMJs induces rapid remodeling of pCaMKII in postsynaptic compartments. (A) Confocal images of NMJs stained with the indicated antibodies at baseline and after PhTx application. **(B)** Quantification of mean intensity showing pCaMKII levels are reduced within 10 min following PhTx application in wild type and *GluRIIA^{Q615R}* NMJs.

Reviewer Figure 3: PhTx application to *GluRIIA^{Q615R}* NMJs induces rapid PHP expression. (A) Schematics and electrophysiological traces of the indicated genotypes and conditions. **(B)** Quantification mEPSP and quantal content values after PhTx application normalized to baseline values. The homeostatic increase in quantal content after PhTx application is observed in *GluRIIA^{Q615R}* mutants.

This new data strongly indicates that the GluRIIA C-tail motif we have identified in this study is both necessary and sufficient activate CaMKII, in the absence of any Ca^{2+} generated by synaptic activity. We thank

both reviewers for encouraging these experiments, which have greatly improved our study.

Other points:

1. It would be helpful for the reader if the authors described the mutants used in Fig. S1, (eg. CaMKII-Ntide, CaMKII-ala, etc) in the figure legend and/or text with the name of the mutant and the associated phenotype (kinase dead etc) so a non-CaMKII expert could understand the data more easily (eg. why pCaMKII decreases in CaMKII-Ntide; CaMKII-ala = does what?).

We appreciate this suggestion by the reviewer and absolutely agree. We have now added a more complete description of the various manipulations of CaMKII in the **Fig. S1 legend**.

2. The cartoons and graphs are a clever and clear way to show complex experiments. For readers that might not be able to distinguish the colors, it would be helpful to also label the bar graphs below each sample or have a legend next to them for extra clarity (eg. Fig 3).

This is also a good point and one in which we fully agree. We have now added labels to the bar graphs below each sample in **Figures 2F, 4B-E, 5B-E, and 8B-E** in the revised manuscript.

RESPONSE TO REVIEWER 2

Reviewer 2 states: Presynaptic homeostatic plasticity is thought to be important for neuronal function. Here Perry et al report that the c-tail of GluRIIA is critical for recruiting activated CaMKII, which mediates PHP signaling. Some results of this manuscript, especially the part showing that c-tail of GluRIIA is important for docking CaMKII, is interesting, and the related experiments were well designed and solid. However. The conclusion regarding whether CaMKII activity is sensitive to reductions of postsynaptic calcium warrants further study. Based on their results, it is very likely that the confirmational changes of c-tail in GluRIIA is important for recruiting activated CaMKII for PHP. If this is the case, however, this result is largely expected as it is exactly the way how postsynaptic CaMKII gets activated in mammals, which unfortunately reduces the significance of this manuscript for publication in Nature Communications.

We thank the reviewer for his/her incisive comments and for encouraging further study to clarify and extend the key findings presented in our original manuscript. We detail below how we have addressed the valid concerns raised by the reviewer, namely regarding **1)** the specificity of the CaMKII/pCaMKII antibodies generated in this study and **2)** whether pCaMKII activation is truly Ca²⁺-independent. However, would first like to briefly respond to the reviewer's point about the significance of our study.

The reviewer is of course correct that CaMKII is activated by a GluR C-tail in concert with Ca²⁺ in mammals. However, this involves the NMDA receptor C-tail, which has no sequence homology or even homologous function to the Drosophila NMJ GluRs, which are classified as kainate-type GluRs (KARs; (Li et al., 2016)). Indeed, as far as we know, CaMKII has never been shown to be recruited or activated by a non-NMDA GluR C-tail. Thus, our study is the *first to show CaMKII activity can be controlled by a non-NMDA GluR*.

Further, we now present strong evidence that the C-tail of the GluRIIA KAR subunit is necessary and sufficient to activate CaMKII, independently of any synaptic Ca²⁺ influx (new **Fig. 3**). Together, we respectfully emphasize that these new insights into the roles of a KAR C-tail and synaptic Ca²⁺ in regulating CaMKII activity were not at all expected. This regulation of CaMKII by KARs in homeostatic

synaptic plasticity differs in novel and unanticipated ways from our established understanding of how CaMKII is regulated by synaptic Ca²⁺ and NMDARs in Hebbian plasticity.

Beyond these significant insights, we anticipate that **1)** overturning a model of homeostatic plasticity induction that has persisted for over 20 years, based on reduced Ca²⁺ influx, and **2)** defining active CaMKII as exerting a constitutive suppression of retrograde signaling at synapses will also prove to be major findings of high impact to the field contributed by our manuscript.

1: The main findings are based on new antibodies developed by the authors. However, RNAi results show that pCaMKII or CaMKII staining only decreased ~50%, which makes me worry about whether these antibodies are specific. The authors should knock out CaMKII in their system and test the specificity of those antibodies. It is also more worrying as the authors saw that CaMKII intensity decreased when they overexpress CaMKII T287D (Fig, S1C, D).

We absolutely agree that it is crucial to ensure that the new pCaMKII and CaMKII antibodies we generated in this study are specific, which is why we dedicated an entire Supplementary Figure (Fig. S1) to evaluating this very question. First, we would like to respectfully emphasize that *every experiment presented in Fig. S1 is consistent with the expected pCaMKII or CaMKII signal if these antibodies were indeed specific.* The CaMKII-RNAi knock down experiment reduced CaMKII by ~50%, and as expected, we observed a similar concomitant reduction in the pCaMKII and CaMKII signal (Fig. S1A-D). While ideally the CaMKII-RNAi would have reduced CaMKII levels even further, it is not clear to us why this result should render the specificity of our antibodies in question.

In addition to CaMKII-RNAi, we also used two inhibitors of CaMKII activity, CaMKII-Ntide (Chang et al., *PNAS* 1998) and CaMKII-Ala (Jin et al., *J. Neurosci* 1998; Joiner and Griffith, *J. Neurosci* 1997; Wang et al., *Neuron* 1994), each of which selectively reduced the anti-pCaMKII signal, as expected, without altering the anti-CaMKII signal (Fig. S1A-D). Similarly, overexpression of CaMKII enhanced the signal of both pCaMKII and CaMKII (Fig. S1A-D), as expected. Together, these multiple and independent lines of evidence demonstrate the specificity of the two new antibodies developed in our study.

The reviewer raises an important point about the change in CaMKII signal following overexpression of constitutively active *CaMKII^{T287D}*. As expected, overexpression of this transgene greatly enhanced the pCaMKII signal (Fig. S1A,B). However, the reviewer is correct that the CaMKII signal was unexpectedly reduced by overexpression of constitutively active *CaMKII^{T287D}* (Fig. S1C,D). However, we do not think this result provides any indication that the pCaMKII or CaMKII antibodies are non-specific. Rather, the reviewer may have also

noticed that the postsynaptic scaffold DLG/PSD95 was also reduced by *CaMKII^{T287D}* overexpression (Fig. S1C,D). Indeed, there appears to be a close relationship between levels of CaMKII and DLG/PSD95, which is not unexpected given the numerous studies highlighting the close link between these two major postsynaptic scaffolds (Kim and Sheng, *Nat Reviews Neuroscience* 2004; Koh et al., *Cell* 1999; Mauceri

Reviewer Figure 5: Postsynaptic DLG knock down also reduces CaMKII levels without impacting pCaMKII. (A) Confocal images of wild-type and DLG-RNAi NMJs showing both DLG and CaMKII immuno-intensity levels are reduced, while pCaMKII levels are unchanged. **(B)** Quantification of DLG, CaMKII, and pCaMKII levels in DLG-RNAi normalized to wild-type values.

et al., *JBC* 2004; Steiner et al., *Neuron* 2008). To further underscore this point, we have stained pCaMKII and CaMKII at NMJs in which we have substantially reduced DLG using a DLG-RNAi in muscle (**Reviewer Fig. 5**). Consistent with the data in Fig. S1C,D, we observed that when DLG levels are reduced (by over 90%), CaMKII levels are similarly reduced (by over 70%), while pCaMKII levels are unchanged (Reviewer Fig. 5). The reason *CaMKII^{T287D}* overexpression reduces DLG levels is unclear, but may be related to DLG being a putative phosphorylation substrate of CaMKII (Koh et al., *Cell* 1999; Steiner et al., *Neuron* 2008). We are pursuing this interesting question in an ongoing project, which will be the subject of a future study. Nonetheless, these results serve to further underscore the specificity of the CaMKII and pCaMKII antibodies.

To further validate the specificity of the pCaMKII and CaMKII antibodies, we have stained NMJs overexpressing an inactive form of CaMKII, *CaMKII^{T287A}*. We find, as expected, pCaMKII levels are significantly reduced while CaMKII levels are unchanged (revised **Fig. S1A-D**). This new data is now presented in a revised Fig. S1, which we hope the reviewer agrees serves as additional evidence of the specificity of the pCaMKII/CaMKII antibodies developed in this study.

Finally, regarding the suggestion by the reviewer to stain pCaMKII/CaMKII in CaMKII KO animals, this is obviously something we like to do. Indeed, towards this goal we have generated specific and unambiguous null mutations in *Drosophila* CaMKII using CRISPR/Cas9 mutagenesis.

Unfortunately, these mutants are embryonic lethal (data not shown), preventing an opportunity to demonstrate the specificity of the pCaMKII/CaMKII antibodies on null mutants in the larval preparation. However, in response

to the reviewer's larger point, we have developed a "tissue-specific KO" using a somatic CRISPR mutagenesis approach to mutate CaMKII specifically in muscle. In particular, we generated a *Cas9* expression cassette under the control of the muscle-specific promoter *Mef2* (**Reviewer Fig. 6**). When crossed with a cassette ubiquitously expressing three sgRNAs targeting CaMKII, the animals did live to third-instar stages, where we observed an over 80% reduction in the CaMKII and pCaMKII signals, while GluRIIA levels were unchanged (Reviewer Fig. 6). We would be happy to include this data in the manuscript if requested by the reviewer, although we feel the many lines of evidence discussed above are sufficiently compelling to highlight the specificity of the new pCaMKII/CaMKII antibodies developed in this study, which we expect to become major tools in the field moving forward.

2. CaMKII activation is a dynamic process, which is known to be triggered by neuronal activities. There are many factors that may affect CaMKII activation, such as the frequency of neuronal firing, calcium influx, and conformational changes of glutamate receptors. Indeed, it has been well known in mammals that calcium is necessary, but not sufficient for inducing postsynaptic CaMKII activation, as activity-dependent conformational changes in NMDARs are also required for this process. Therefore, the authors should examine the frequency of postsynaptic calcium transients to see whether it is affected by their genetic manipulations and fix the time point for measuring pCaMKII for each neuronal activation. Moreover, reducing ~50% calcium does not

Reviewer Figure 6: Muscle-specific CaMKII mutagenesis reduces CaMKII levels by over 80%. (A) Confocal images of wild-type and *Mef2>Cas9+CaMKII* sgRNAs NMJs immunostained with CaMKII, pCaMKII, and GluRIIA. As expected, both CaMKII and pCaMKII show similar reductions of >80%, while GluRIIA levels are unchanged. **(B)** Quantification of CaMKII, pCaMKII, and GluRIIA levels in CaMKII-KO muscle normalized to wild-type values.

mean too much, as it is likely that the 50% calcium is still enough to trigger CaMKII activation. They may use calcium chelators such as BAPTA-AM to buffer all calcium influx and examine whether CaMKII can be still activated.

We greatly appreciate these points raised by the reviewer, and we have spent considerable time working to address the specific comments regarding whether CaMKII activation is Ca²⁺-dependent. The central question really revolves around whether CaMKII is activated independently of Ca²⁺ in the context of homeostatic plasticity or, as is well known in mammalian synapses, Ca²⁺ is necessary along with conformational changes in the GluR C-tail due to glutamate binding. We agree that in the original manuscript, we did not completely rule out Ca²⁺ being necessary for CaMKII activation since, as the reviewer correctly points out, there is still ~50% residual Ca²⁺ signaling in the absence of GluRIIA or Ca²⁺ permeability through this receptor, presumably due to residual activity of GluRB receptors.

To address this key outstanding question, we have developed a new tool, presented in an entirely new Fig. 3. Specifically, we have employed a botulinum neurotoxin (BoNT-C) that, when expressed in motor neurons, eliminates miniature and evoked synaptic activity (new Fig. 3A). When BoNT-C is expressed in all motor neurons, the animals die as embryos, as expected, because there is no glutamate release or neurotransmission required for muscle contraction. However, when BoNT-C is expressed in a subset of motor neurons using *OK319-Gal4*, which drives *Gal4* expression in the motor neurons that innervate muscles 6/7 and 4 (Sweeney et al., *Neuron* 1995), embryonic lethality is circumvented. *This enables an investigation of CaMKII activity at NMJs devoid of glutamate release and any conformational signaling or synaptic Ca²⁺ influx.* We first show that BoNT-C expression eliminates synaptic transmission (Fig. 3A). Next, we developed and now employ the newest and most sensitive GCaMP variant, GCaMP8f, to confirm the absence of any Ca²⁺ influx at BoNT-C NMJs (Fig. 3B). Finally, we show that at NMJs devoid of any glutamate release or postsynaptic Ca²⁺ influx, pCaMKII levels are unchanged from wild type (Fig. 3C,D). Finally, pCaMKII levels continue to be abolished in the absence of *GluRIIA* (Fig. 3C,D), as expected. This new data is perhaps the most important in the revised manuscript, which *clearly and unequivocally demonstrates the novel mode of regulation we have revealed in this study, where pCaMKII levels are dependent entirely on the physical presence of the GluRIIA C-tail, independently of glutamate-induced conformational signaling or synaptic Ca²⁺ influx.* We thank the reviewer for pushing us to provide this necessary and compelling data.

As suggested by the reviewer, we have also added a new Supplementary Figure (**Fig. S4**), where we show that the patterns of evoked synaptic activity between wild type and *GluRIIA* mutant NMJs are unchanged in vivo (Fig. S4A,B), and further that acute application of the Ca²⁺ chelator BAPTA-AM to NMJs does not lead to major reductions in pCaMKII levels (Fig. S4C,D). Together, this combination of new data clearly demonstrates a novel mechanism for regulating CaMKII activity, where a KAR C-tail is sufficient to fully activate pCaMKII levels in the absence of glutamate-induced conformational signaling or synaptic Ca²⁺ influx.

REFERENCES

Bayer, K.U., de Koninck, P., Leonard, A.S., Hell, J.W., and Schulman, H. (2001). Interaction with the NMDA receptor locks CaMKII in an active conformation. *Nature* 411, 801–805.

Chang, B.H., Mukherji, S., and Soderling, T.R. (1998). Characterization of a calmodulin kinase II inhibitor protein in brain. *Proceedings of the National Academy of Sciences* 95.

- Frank, C.A., Kennedy, M.J., Goold, C.P.P., Marek, K.W., and Davis, G.W.W. (2006). Mechanisms Underlying the Rapid Induction and Sustained Expression of Synaptic Homeostasis. *Neuron* 52, 663–677.
- Jin, P., Griffith, L.C., and Murphey, R.K. (1998). Presynaptic Calcium/Calmodulin-Dependent Protein Kinase II Regulates Habituation of a Simple Reflex in Adult *Drosophila*. *Journal of Neuroscience* 18, 8955–8964.
- Joiner, M.A., and Griffith, L.C. (1997). CaM Kinase II and Visual Input Modulate Memory Formation in the Neuronal Circuit Controlling Courtship Conditioning. *Journal of Neuroscience* 17, 9384–9391.
- Kim, E., and Sheng, M. (2004). PDZ domain proteins of synapses. *Nature Reviews Neuroscience* 2004 5:10 5, 771–781.
- Kiragasi, B., Wondolowski, J., Li, Y., and Dickman, D.K. (2017). A Presynaptic Glutamate Receptor Subunit Confers Robustness to Neurotransmission and Homeostatic Potentiation. *Cell Reports* 19, 2694–2706.
- Koh, Y.H., Popova, E., Thomas, U., Griffith, L.C., and Budnik, V. (1999). Regulation of DLG Localization at Synapses by CaMKII-Dependent Phosphorylation. *Cell* 98, 353–363.
- Li, Y., Dharkar, P., Han, T.H., Serpe, M., Lee, C.H., and Mayer, M.L. (2016). Novel Functional Properties of *Drosophila* CNS Glutamate Receptors. *Neuron* 92, 1036–1048.
- Mauceri, D., Cattabeni, F., Luca, M. di, and Gardoni, F. (2004). Calcium/Calmodulin-dependent Protein Kinase II Phosphorylation Drives Synapse-associated Protein 97 into Spines *. *Journal of Biological Chemistry* 279, 23813–23821.
- Sanhueza, M., and Lisman, J. (2013). The CaMKII/NMDAR complex as a molecular memory. *Molecular Brain* 6, 1–8.
- Steiner, P., Higley, M.J., Xu, W., Czervionke, B.L., Malenka, R.C., and Sabatini, B.L. (2008). Destabilization of the Postsynaptic Density by PSD-95 Serine 73 Phosphorylation Inhibits Spine Growth and Synaptic Plasticity. *Neuron* 60, 788–802.
- Sweeney, S.T., Broadie, K., Keane, J., Niemann, H., and O’Kane, C.J. (1995). Targeted expression of tetanus toxin light chain in *Drosophila* specifically eliminates synaptic transmission and causes behavioral defects. *Neuron* 14, 341–351.
- Wang, J., Renger, J.J., Griffith, L.C., Greenspan, R.J., and Wu, C.F. (1994). Concomitant alterations of physiological and developmental plasticity in *drosophila* CaM kinase II-inhibited synapses. *Neuron* 13, 1373–1384.

Reviewers' comments:

Reviewer #1 (Remarks to the Author):

The authors have responded to my comments and the paper is now appropriate for publication in Nature Communications.

Reviewer #2 (Remarks to the Author):

In their revised manuscript, Perry et al., have done some experiments to touch the questions that I have raised previously, but did not really address most of them.

1: The authors should examine whether NMDARs are affected when they manipulated GluRIIA, including the localization, the expression, and the function of NMDARs.

2: They authors should show the efficiency of RNAi knock down, including the mRNA level and the protein level. If RNAi knocked down the expression of CaMKII more than ~50%, then we should expect that the reduction of CaMKII in their Fig. S1 should be more than ~50%. Now, it is very difficult to interpretate their results, as I am not sure whether the ~50% reduction they have observed is due to the nonspecific staining of CaMKII antibody or is due to the efficiency of their RNAi.

3: The tissue-specific KO is helpful but may also raise more questions. A: It is known that the specificity of CaMKII antibody differs in different tissues; B: the isoforms of CaMKII may be different in different tissues. Although CaMKII KO animals are lethal, they authors can use a conditional KO system to KO CaMKII when the animals become mature.

4: I am surprised that the authors did not use the other methods to validate their results, including mRNA and the WB. This is somehow unusual, especially considering that the antibody issue is critical for this paper. Simply staining the NMJ boutons is not enough, as the staining signal can come from the change of the protein localization. For example, it is known that the phosphorylation of CaMKII (287D) can affect the synaptic localization of CaMKII.

5: It is true that there is a relationship between DLG/PSD-95 and CaMKII. However, I believe that this refers to the postsynaptic expression of CaMKII, not the total level of CaMKII in cells. It is difficult to understand why the authors think this relationship can help for explaining the specificity of the antibody. It is possible that their pCaMKII antibody can work, but whether the CaMKII antibody works requires more experiments to prove. For example, how do we know whether the target of their CaMKII antibody is CaMKII, but not DLG/PSD95?

6: The newly added figure 3 nicely show that in the absence of synaptic transmission, deleting the GluRIIA C-tail impaired the phosphorylation of CaMKII at 287. This experiment provides a piece of evidence suggesting that the GluRIIA C-tail is necessary for pCaMKII in the absence of synaptic activity but did not exclude the necessity of calcium in recruiting pCaMKII under a physiological condition in WT animals. BAPTA-AM is a telling experiment, but as shown by the authors, BAPTA-AM significantly reduce pCaMKII, suggesting that calcium is required for pCaMKII. Also, the authors should show data for comparing spontaneous but not evoked synaptic activity between WT and the mutated NMJs, which more likely reflects endogenous synaptic activity of neurons that may regulate pCaMKII.

7: As the activation of CaMKII is a dynamic process, it is hard to agree that phosphorylation state of CaMKII at 287 really indicates the activation of CaMKII (which is determined by K42/K43 as a kinase). Normally, people will do these kind experiments in vitro, in which the stimulating parameters can be precisely controlled (e.g. the fixation time following the stimulation). In this manuscript, there are so many factors that may be affected by their genetic manipulated flies, which makes their findings unclear. Moreover, pCaMKII is not an accurate indicator of the CaMKII activity, as it mainly reflects

the autophosphorylation of CaMKII, but not really the activation of CaMKII. Thus, the pCaMKII in this manuscript may more likely reflect the state of the autophosphorylation of CaMKII. Maybe monitoring the activation of CaMKII using a CaMKII sensor and a calcium sensor at the same time can be helpful. Also, it will be also helpful to examine some known substrates of CaMKII, to make sure that pCaMKII really indicates the activation of CaMKII in their results.

Reviewer #3 (Remarks to the Author):

In this study, Perry et al investigated the postsynaptic mechanisms responsible for the induction of PHP in the *Drosophila* NMJ. Using a series of well-designed and carefully executed experiments, the authors identified the loss of C-terminal tail of GluRIIA as the signal that leads to retrograde PHP induction, while disproving the involvement of reduced calcium signaling in this process.

To directly tested the long-standing model and separate the effects generated by reduced postsynaptic calcium influx and the diminished GluRs, the authors developed several new reagents, including specific antibodies for total and phosphorylated-CAMKII, new mutant alleles of GluIIA, a postsynaptic GCaMP indicator, and a BoNT-C transgene to block synaptic vesicle release. These impressive efforts produced strong experimental evidence that support a calcium independent mechanism regulating postsynaptic CaMKII and homeostatic plasticity, an interesting and significant finding that meet the high publication standard of Nature Communications.

The manuscript is well written and easy to follow. Figures are also nice and clear. Previous reviewers' concerns have been adequately addressed during the revision. I have no additional concerns and would support the manuscript's publication.

RESPONSE TO REVIEWS

We thank the Editor and the three Reviewers for their time and efforts in evaluating our manuscript and for providing constructive comments and criticisms. We are further encouraged by the positive comments from the new Reviewer 3, who, along with Reviewer 1, deems the revised manuscript to be appropriate for publication in *Nature Communications*.

There were lingering concerns expressed by Reviewer 2 about the specificity of the CaMKII antibodies we generated in this study. Reviewer 2 suggested additional experiments that were not proposed in his/her initial review. We have therefore spent the past five months focused on performing new experiments in response to Reviewer 2's suggestions in this revised manuscript.

As detailed below, we now include several new experiments to respond the additional concerns raised by Reviewer 2. These include new characterizations of NMDAR mutants that we generated using CRISPR/Cas9 gene editing and, importantly, immunoblots against CaMKII and DLG in various genetic conditions. In short, these experiments demonstrate:

- 1) NMDARs in *Drosophila* are not expressed in the postsynaptic muscle and do not impact postsynaptic CaMKII activity or PHP expression. This data is now included in a new **Supplementary Figure 6**.
- 2) Immunoblots further demonstrate that the CaMKII antibody is specific, recognizing a single band at the expected molecular mass (~60 kDa). This band is reduced by ~50% in CaMKII RNAi, matching what we observed by immunostaining, and further, rules out any alternative CaMKII splice isoforms functioning in the muscle. Finally, we show that the CaMKII antibody does not cross react with DLG, which runs as two isoforms at ~110 and ~120 kDa. All of this data is now presented in an entirely new **Supplementary Figure 2**.

These two new supplementary figures are included along with additional revisions to the text (highlighted in gray).

Together, these new experiments further establish the specificity of the CaMKII antibodies generated in this study and highlight the unique role of *Drosophila* kainate-type glutamate receptors in controlling CaMKII activity to gate the retrograde induction of presynaptic homeostatic plasticity.

RESPONSE TO REVIEWER 1

Reviewer 1 states: *The authors have responded to my comments and the paper is now appropriate for publication in Nature Communications.*

We are grateful to the reviewer for his/her positive assessment of our study.

RESPONSE TO REVIEWER 2

Reviewer 2 states: *In their revised manuscript, Perry et al., have done some experiments to touch the questions that I have raised previously, but did not really address most of them.*

We thank the reviewer for recognizing the efforts made to address the questions raised in their previous review. As detailed below, we have dedicated the past five months towards further addressing the additional concerned raised.

1: The authors should examine whether NMDARs are affected when they manipulated *GluRIIA*, including the localization, the expression, and the function of NMDARs.

This is an interesting question given that *GluRIIA* is a kainate-type receptor subunit, a glutamate receptor subtype that to our knowledge has never been shown to directly interact with or modulate CaMKII activity. As the Reviewer knows, this ability has only been shown in mammals in the case of NMDARs. Although this question about *Drosophila* NMDARs was not raised in Reviewer 2's original critique of our manuscript, we have addressed this important point with entirely new data now added to the revised manuscript.

There are two NMDA receptors encoded in the *Drosophila* genome, NMDAR1 and NMDAR2. To address the reviewer's questions, we have done the following:

1) We have obtained T2A-GAL4 knock-in of both NMDAR1 and 2 to determine expression of these two receptors in third-instar larvae. We find that for both NMDARs, strong expression is observed in the larval brain as well as in larval motor neurons, where they are likely located in dendrites to receive glutamatergic signaling from pre-motor neurons (Li et al., 2016; Rohrbough and Broadie, 2002). However, we find that neither NMDAR is expressed in the postsynaptic muscle, where PHP inductive signaling and CaMKII activity are studied in this manuscript. This data is shown in the new **Supplementary Figure 6B**.

2) Further, we have used CRISPR mutagenesis to generate null mutations in both NMDAR1 and NMDAR2. We find no defects in baseline electrophysiology in either mutant. In addition, robust PHP is expressed in *NMDAR, GluRIIA* double mutants. This data is now shown in the new **Supplementary Figure 6A,C,D**.

3) Finally, we have immunostained CaMKII and pCaMKII at NMDAR mutant NMJs. As expected, we find no significant change in CaMKII or pCaMKII signals. This data is shown in **Supplementary Figure 6E,F**.

Together, this important data rules out any contribution or impact of NMDARs on CaMKII activity in postsynaptic compartments of the NMJ at baseline and in *GluRIIA* mutants, further highlighting the unique ability of the *GluRIIA* kainate receptor subunit to control CaMKII activity and PHP expression demonstrated in our study.

2: The authors should show the efficiency of RNAi knock down, including the mRNA level and the protein level. If RNAi knocked down the expression of CaMKII more than ~50%, then we should expect that the reduction of CaMKII in their Fig. S1 should be more than ~50%. Now, it is very difficult to interpretate their results, as I am not sure whether the ~50% reduction they have observed is due to the nonspecific staining of CaMKII antibody or is due to the efficiency of their RNAi.

We understand this point and have now performed immunoblot analysis to further support the specificity of the CaMKII antibodies generated in our study. First, we note that it is not uncommon for RNAi knock down to reduce protein levels encoded by the target gene by 50% or less, particularly in *Drosophila* larval stages, where time is limited for the efficiency of RNAi knock down and perdurance of the protein may persist. Indeed, numerous papers have demonstrated similar reductions of protein after RNAi knock down (Dietzl et al., 2007; Malik et al., 2013). Therefore, the results reported in **Supplementary Figure S1**, in which CaMKII levels are reduced by ~50% by immunostaining, are not at all surprising.

Second, mRNA levels can poorly correlate with protein levels, at least in *Drosophila*, as there are many aspects of post-transcriptional regulation that do not correlate with mRNA expression but that have major

impacts on actual protein synthesis and steady-state protein levels, which is ultimately what one cares about. Indeed, we previously developed a ribosome profiling technology to demonstrate this very point (Chen and Dickman, 2017). Specifically, we found that mRNA transcriptional levels (assessed by qPCR, which appears to be what the Reviewer is suggesting), do not at all correlate well with protein levels in the third-instar muscle at the *Drosophila* NMJ, and have no reliable quantitative prediction with actual protein levels in basal states or even after RNAi knock down. Thus, quantifying mRNA levels after RNAi knock down and testing for quantitative correlation with immunostaining levels would not address the reviewer's concern.

Therefore, we have performed Western blot analysis of CaMKII levels in muscle lysates from wild type, CaMKII-RNAi, and CaMKII overexpression. Importantly, this new analysis shows that our CaMKII antibody recognizes a single band at the expected molecular mass of *Drosophila* CaMKII (~60 kDa; see new **Supplementary Figure S2B**). Second, RNAi knock down reduces the intensity of CaMKII protein by a similar degree as we observed via immunostaining (see **Supplementary Figure S2C,D**). These results entirely support the specificity of the CaMKII antibody generated in our study.

This new data is presented in **Supplementary Figure S2** and associated text in the revised manuscript. This new data further supports and validates the specificity of the CaMKII antibody generated in our study.

3: The tissue-specific KO is helpful but may also raise more questions. A: It is known that the specificity of CaMKII antibody differs in different tissues; B: the isoforms of CaMKII may be different in different tissues. Although CaMKII KO animals are lethal, they authors can use a conditional KO system to KO CaMKII when the animals become mature.

We respond to these points with the following:

A. There is no evidence to suggest the specificity of the anti-*Drosophila* CaMKII antibodies we generated in this study differs in different tissues.

B. Although previous studies have found evidence that up to three *Drosophila* CaMKII isoforms may be expressed in the adult fly brain (Griffith and Greenspan, 1993; Guptaroy et al., 1996; Ohsako et al., 1993), the immunoblot analysis discussed in Point 2 above and shown in the new **Supplemental Figure S2** shows that our CaMKII antibody recognizes a single band in larval muscle lysates. Importantly, the mass of this band matches the expected molecular mass of CaMKII (~60 kDa), and the intensity of this band decreases with CaMKII-RNAi and increases with CaMKII overexpression. This new data strongly suggests only a single isoform is expressed in the tissue relevant to the study, larval muscle. Furthermore, the RNAi sequence targets common amino acids present in all possible CaMKII isoforms. Similarly, the CaMKII amino acid sequence used to generate our CaMKII antibody is also present in every possible CaMKII splice isoform. Thus, a single CaMKII isoform is expressed in *Drosophila* larval muscle and is reliably targeted by the reagents used in our study.

C. Conditional KO systems (e.g. genes or exons flanked by FLP recombinases) have recently been developed in *Drosophila* but require extensive time (months-years) for the genetic engineering and crosses necessary to successfully implement and validate. More importantly, these systems do not work to remove all protein at larval stages in our experience and as reported from other labs in the field. For example, significant vesicular glutamate transporter protein (vGlut) persists in third-instar larvae after conditional KO (Han et al., 2022), as does the Cav2 Ca²⁺ channel Cac after conditional Cac KO (Cunningham et al., 2022). Moreover, CaMKII null larvae still show significant CaMKII immunostaining, likely due to maternally contributed mRNA (Kuklin et al., 2017). Thus, a conditional KO approach would

not be more effective than the multiple other approaches we have used in our study to remove CaMKII protein in larval muscle.

4: I am surprised that the authors did not use the other methods to validate their results, including mRNA and the WB. This is somehow unusual, especially considering that the antibody issue is critical for this paper. Simply staining the NMJ boutons is not enough, as the staining signal can come from the change of the protein localization. For example, it is known that the phosphorylation of CaMKII (287D) can affect the synaptic localization of CaMKII.

This issue was addressed in Points 2 and 3 above. mRNA is not an accurate or quantitative predictor of protein levels for a variety of reasons articulated above and demonstrated in Chen and Dickman, 2017. We have now performed Western blot analysis of CaMKII in knock down and overexpression, and each of these results strongly support the specificity of the CaMKII antibody generated in our study. We have spent considerable time and effort to optimize immunoblot analysis of the pCaMKII antibody generated in our study. Unfortunately, we were unable to obtain any signal on a Western blot with this phospho-specific antibody which, in our hands and others (e.g. Karen Chang, USC, *personal communication*), is not uncommon for phospho-specific antibodies.

5: It is true that there is a relationship between DLG/PSD-95 and CaMKII. However, I believe that this refers to the postsynaptic expression of CaMKII, not the total level of CaMKII in cells. It is difficult to understand why the authors think this relationship can help for explaining the specificity of the antibody. It is possible that their pCaMKII antibody can work, but whether the CaMKII antibody works requires more experiments to prove. For example, how do we know whether the target of their CaMKII antibody is CaMKII, but not DLG/PSD95?

The reviewer is correct that the pCaMKII signal does not at all overlap with anti-DLG, but that the CaMKII signal does largely overlap with DLG in the postsynaptic compartment. In the previous revision and response to reviewer, we showed that RNAi knock down of CaMKII reduced total CaMKII immunostaining but not DLG immunostaining levels, while RNAi knock down of DLG reduced DLG immunostaining but not CaMKII (previous Response to Reviewers Figure and **Figure S1C,D**). These results already provide strong evidence that the CaMKII antibody recognizes CaMKII and does not cross react with DLG.

To further demonstrate the specificity of the CaMKII antibody, and to provide more clarity about the relationship between DLG and CaMKII, we have now performed Western blot analysis of CaMKII and DLG antibodies validated in previous studies (Koh et al., 1999; Lahey et al., 1994; Mendoza-Topaz et al., 2008; Woods et al., 1996). Importantly, we find that the anti-CaMKII antibody recognizes a single band that runs at the expected CaMKII molecular mass (~60 kDa), and this band changes in the expected ways with CaMKII RNAi or overexpression (**Supplemental Figure S2B-D**). In contrast, the anti-DLG immunoblot recognizes two bands, running at ~110 and ~120 kDa, corresponding to the expected DLG-A and DLG-S97 isoforms (**Supplemental Figure S2A**), consistent with previous studies. Similar to our immunostaining data discussed above, this new immunoblot analysis clearly shows no cross reactivity between anti-CaMKII and anti-DLG. This data is now shown in the new **Supplemental Figure S2A** and discussed in the associated text of the revised manuscript.

6: The newly added figure 3 nicely show that in the absence of synaptic transmission, deleting the GluRIIA C-tail impaired the phosphorylation of CaMKII at 287. This experiment provides a piece of evidence suggesting that the GluRIIA C-tail is necessary for pCaMKII in the absence of synaptic activity but did not exclude the necessity of calcium in recruiting pCaMKII under a physiological condition in WT animals. BAPTA-AM is a telling experiment, but as shown by the authors, BAPTA-AM significantly reduce pCaMKII, suggesting that calcium is required for pCaMKII. Also, the authors should show data for comparing spontaneous but not evoked synaptic activity between

WT and the mutated NMJs, which more likely reflects endogenous synaptic activity of neurons that may regulate pCaMKII.

It is well established that in both mammalian and *Drosophila* systems, Ca^{2+} can activate CaMKII (Griffith, 2004; Hell, 2014; Herring and Nicoll, 2016; Hodge et al., 2006; Thornquist et al., 2020), and we did not intend to imply that the unique role of the GluRIIA C-tail we have uncovered in this study is the only control mechanism that can activate CaMKII. Rather, in the absence of synaptic activity (as the Reviewer points out was shown in the new Figure 3), the GluRIIA C-tail is sufficient to maintain activated, “autonomous” CaMKII activity. Clearly Ca^{2+} signaling could serve to further modulate this activity (as may be suggested by the BAPTA-AM experiment the Reviewer refers to), but in terms of the central question being addressed in our study, Ca^{2+} signaling is not the necessary inductive signal that modulates CaMKII activity or the expression of retrograde homeostatic plasticity.

It is not clear to us what experiment the Reviewer is suggesting in asking for “*comparing spontaneous but not evoked synaptic activity between WT and mutated NMJs*”. We have shown Ca^{2+} imaging of spontaneous release between WT and *GluRIIA* mutant NMJs (in addition to several other genotypes) in **Figure 2D,F**, as well as endogenously evoked Ca^{2+} imaging shown in **Supplemental Figure S5**. Electrophysiological data for spontaneous activity (mEPSP amplitude, frequency) is shown for all genotypes in this study in **Figure 1B,C; Figure 4A,B; Figure 6A,B; Figure 8A,C; and Supplementary Table S1**.

7: As the activation of CaMKII is a dynamic process, it is hard to agree that phosphorylation state of CaMKII at 287 really indicates the activation of CaMKII (which is determined by K42/K43 as a kinase). Normally, people will do these kind experiments in vitro, in which the stimulating parameters can be precisely controlled (e.g. the fixation time following the stimulation). In this manuscript, there are so many factors that may be affected by their genetic manipulated flies, which makes their findings unclear. Moreover, pCaMKII is not an accurate indicator of the CaMKII activity, as it mainly reflects the autophosphorylation of CaMKII, but not really the activation of CaMKII. Thus, the pCaMKII in this manuscript may more likely reflect the state of the autophosphorylation of CaMKII. Maybe monitoring the activation of CaMKII using a CaMKII sensor and a calcium sensor at the same time can be helpful. Also, it will be also helpful to examine some known substrates of CaMKII, to make sure that pCaMKII really indicates the activation of CaMKII in their results.

It is well established in both *Drosophila* and mammalian systems that the phosphorylation state of CaMKII T287 (T286 in mammals) is a sensitive indicator of the activity state of the CaMKII holoenzyme. In *Drosophila*, many studies have demonstrated the intimate relationship between CaMKII T287 phosphorylation and CaMKII activity, both in vitro and in vivo (Haghighi et al., 2003; Park et al., 2002; Sun et al., 2004; Wang et al., 1998). Perhaps most compellingly, phospho-mimetic CaMKII mutations at T287 (i.e. CaMKII^{T287D}) in *Drosophila* induce constitutively active CaMKII signaling in vivo and in vitro, while Alanine mutations at the same residue (CaMKII^{T287A}) render CaMKII constitutively inactive (Haghighi et al., 2003; Park et al., 2002; Vonhoff and Keshishian, 2017; Wang et al., 1998). Indeed, these properties of CaMKII activity have been validated on known substrates of *Drosophila* CaMKII in vitro and in vivo (Koh et al., 1999; Wang et al., 2002). We agree with the Reviewer that we could have done a better job of discussing these results in our previous revision, and have now made these points clear in our revised manuscript.

RESPONSE TO REVIEWER 3

Reviewer 3 states: *In this study, Perry et al investigated the postsynaptic mechanisms responsible for the induction of PHP in the Drosophila NMJ. Using a series of well-designed and carefully executed experiments, the authors identified the loss of C-terminal tail of GluRIIA as the signal that leads to retrograde PHP induction, while disproving the involvement of reduced calcium signaling in this process.*

To directly tested the long-standing model and separate the effects generated by reduced postsynaptic calcium influx and the diminished GluRs, the authors developed several new reagents, including specific antibodies for total and phosphorylated-CAMKII, new mutant alleles of GluIIA, a postsynaptic GCaMP indicator, and a BoNT-C transgene to block synaptic vesicle release. These impressive efforts produced strong experimental evidence that support a calcium independent mechanism regulating postsynaptic CaMKII and homeostatic plasticity, an interesting and significant finding that meet the high publication standard of Nature Communications.

The manuscript is well written and easy to follow. Figures are also nice and clear. Previous reviewers' concerns have been adequately addressed during the revision. I have no additional concerns and would support the manuscript's publication.

We are grateful to the reviewer for his/her positive assessment of our study, and for providing important context for the central conclusions conveyed in the revised manuscript.

REFERENCES

- Chen, X., and Dickman, D. (2017). Development of a tissue-specific ribosome profiling approach in *Drosophila* enables genome-wide evaluation of translational adaptations. *PLoS Genet* 13. <https://doi.org/10.1371/JOURNAL.PGEN.1007117>.
- Cunningham, K.L., Sauvola, C.W., Tavana, S., and Littleton, J.T. (2022). Regulation of presynaptic Ca²⁺ channel abundance at active zones through a balance of delivery and turnover. *BioRxiv* 2022.03.14.484315. <https://doi.org/10.1101/2022.03.14.484315>.
- Dietzl, G., Chen, D., Schnorrer, F., Su, K.C., Barinova, Y., Fellner, M., Gasser, B., Kinsey, K., Oettel, S., Scheiblauer, S., et al. (2007). A genome-wide transgenic RNAi library for conditional gene inactivation in *Drosophila*. *Nature* 2007 448:7150 448, 151–156. <https://doi.org/10.1038/nature05954>.
- Griffith, L.C. (2004). Regulation of Calcium/Calmodulin-Dependent Protein Kinase II Activation by Intramolecular and Intermolecular Interactions. *Journal of Neuroscience* 24, 8394–8398. <https://doi.org/10.1523/JNEUROSCI.3604-04.2004>.
- Griffith, L.C., and Greenspan, R.J. (1993). The Diversity of Calcium/Calmodulin-Dependent Protein Kinase II Isoforms in *Drosophila* Is Generated by Alternative Splicing of a Single Gene. *Journal of Neurochemistry* 61, 1534–1537. <https://doi.org/10.1111/J.1471-4159.1993.TB13650.X>.
- Guptaroy, B., Beckingham, K., and Griffith, L.C. (1996). Functional Diversity of Alternatively Spliced Isoforms of *Drosophila* Ca²⁺/Calmodulin-dependent Protein Kinase II: A ROLE FOR THE VARIABLE DOMAIN IN ACTIVATION. *Journal of Biological Chemistry* 271, 19846–19851. <https://doi.org/10.1074/JBC.271.33.19846>.

- Haghighi, A.P., McCabe, B.D., Fetter, R.D., Palmer, J.E., Hom, S., and Goodman, C.S. (2003). Retrograde control of synaptic transmission by postsynaptic CaMKII at the *Drosophila* neuromuscular junction. *Neuron* 39, 255–267. [https://doi.org/10.1016/S0896-6273\(03\)00427-6](https://doi.org/10.1016/S0896-6273(03)00427-6).
- Han, Y., Chien, C., Goel, P., He, K., Pinales, C., Buser, C., and Dickman, D. (2022). Botulinum neurotoxin accurately separates tonic vs phasic transmission and reveals heterosynaptic plasticity rules in *Drosophila*. *BioRxiv* 2022.02.15.480429. <https://doi.org/10.1101/2022.02.15.480429>.
- Hell, J.W. (2014). CaMKII: Claiming center stage in postsynaptic function and organization. *Neuron* 81, 249–265. <https://doi.org/10.1016/j.neuron.2013.12.024>.
- Herring, B.E., and Nicoll, R.A. (2016). Long-Term Potentiation: From CaMKII to AMPA Receptor Trafficking. *Annual Review of Physiology* 78, 351–365. <https://doi.org/10.1146/annurev-physiol-021014-071753>.
- Hodge, J.J.L., Mullasseril, P., and Griffith, L.C. (2006). Activity-Dependent Gating of CaMKII Autonomous Activity by *Drosophila* CASK. *Neuron* 51, 327–337. <https://doi.org/10.1016/j.neuron.2006.06.020>.
- Koh, Y.H., Popova, E., Thomas, U., Griffith, L.C., and Budnik, V. (1999). Regulation of DLG Localization at Synapses by CaMKII-Dependent Phosphorylation. *Cell* 98, 353–363. [https://doi.org/10.1016/S0092-8674\(00\)81964-9](https://doi.org/10.1016/S0092-8674(00)81964-9).
- Kuklin, E.A., Alkins, S., Bakthavachalu, B., Genco, M.C., Sudhakaran, I., Raghavan, K.V., Ramaswami, M., and Griffith, L.C. (2017). The Long 3'UTR mRNA of CaMKII Is Essential for Translation-Dependent Plasticity of Spontaneous Release in *Drosophila melanogaster*. *Journal of Neuroscience* 37, 10554–10566. <https://doi.org/10.1523/JNEUROSCI.1313-17.2017>.
- Lahey, T., Gorczyca, M., Jia, X.X., and Budnik, V. (1994). The *drosophila* tumor suppressor gene *dlg* is required for normal synaptic bouton structure. *Neuron* 13, 823–835. [https://doi.org/10.1016/0896-6273\(94\)90249-6](https://doi.org/10.1016/0896-6273(94)90249-6).
- Li, Y., Dharkar, P., Han, T.H., Serpe, M., Lee, C.H., and Mayer, M.L. (2016). Novel Functional Properties of *Drosophila* CNS Glutamate Receptors. *Neuron* 92, 1036–1048. <https://doi.org/10.1016/j.neuron.2016.10.058>.
- Malik, B.R., Gillespie, J.M., and Hodge, J.J.L. (2013). CASK and CaMKII function in the mushroom body α “ β ” neurons during *Drosophila* memory formation. *Frontiers in Neural Circuits* 0, 52. <https://doi.org/10.3389/FNCIR.2013.00052/BIBTEX>.
- Mendoza-Topaz, C., Urrea, F., Barría, R., Albornoz, V., Ugalde, D., Thomas, U., Gundelfinger, E.D., Delgado, R., Kukuljan, M., Sanxaridis, P.D., et al. (2008). DLGS97/SAP97 Is Developmentally Upregulated and Is Required for Complex Adult Behaviors and Synapse Morphology and Function. *Journal of Neuroscience* 28, 304–314. <https://doi.org/10.1523/JNEUROSCI.4395-07.2008>.
- Ohsako, S., Nishida, Y., Ryo, H., and Yamauchi, T. (1993). Molecular characterization and expression of the *Drosophila* Ca²⁺/calmodulin-dependent protein kinase II gene. Identification of four forms of the enzyme generated from a single gene by alternative splicing. *Journal of Biological Chemistry* 268, 2052–2062. [https://doi.org/10.1016/S0021-9258\(18\)53961-2](https://doi.org/10.1016/S0021-9258(18)53961-2).
- Park, D., Coleman, M.J., Hodge, J.J.L., Budnik, V., and Griffith, L.C. (2002). Regulation of neuronal excitability in *Drosophila* by constitutively active CaMKII. *Journal of Neurobiology* 52, 24–42. <https://doi.org/10.1002/NEU.10066>.

- Rohrbough, J., and Broadie, K. (2002). Electrophysiological analysis of synaptic transmission in central neurons of *Drosophila* larvae. *Journal of Neurophysiology* 88, 847–860. <https://doi.org/10.1152/JN.2002.88.2.847/ASSET/IMAGES/LARGE/9K0822481007.JPEG>.
- Sun, X.X., Hodge, J.J.L., Zhou, Y., Nguyen, M., and Griffith, L.C. (2004). The eag Potassium Channel Binds and Locally Activates Calcium/Calmodulin-dependent Protein Kinase II. *Journal of Biological Chemistry* 279, 10206–10214. <https://doi.org/10.1074/jbc.M310728200>.
- Thornquist, S.C., Langer, K., Zhang, S.X., Rogulja, D., and Crickmore, M.A. (2020). CaMKII Measures the Passage of Time to Coordinate Behavior and Motivational State. *Neuron* 105, 334-345.e9. <https://doi.org/10.1016/J.NEURON.2019.10.018>.
- Vonhoff, F., and Keshishian, H. (2017). In Vivo Calcium Signaling during Synaptic Refinement at the *Drosophila* Neuromuscular Junction. *Journal of Neuroscience* 37, 5511–5526. <https://doi.org/10.1523/JNEUROSCI.2922-16.2017>.
- Wang, Z., Palmer, G., and Griffith, L.C. (1998). Regulation of *Drosophila* Ca²⁺/Calmodulin-Dependent Protein Kinase II by Autophosphorylation Analyzed by Site-Directed Mutagenesis. *Journal of Neurochemistry* 71, 378–387. <https://doi.org/10.1046/J.1471-4159.1998.71010378.X>.
- Wang, Z., Wilson, G.F., and Griffith, L.C. (2002). Calcium/Calmodulin-dependent Protein Kinase II Phosphorylates and Regulates the *Drosophila* Eag Potassium Channel *. *Journal of Biological Chemistry* 277, 24022–24029. <https://doi.org/10.1074/JBC.M201949200>.
- Woods, D.F., Hough, C., Peel, D., Callaini, G., and Bryant, P.J. (1996). Dlg protein is required for junction structure, cell polarity, and proliferation control in *Drosophila* epithelia. *Journal of Cell Biology* 134, 1469–1482. <https://doi.org/10.1083/JCB.134.6.1469>.

Reviewers' comments:

Reviewer #2 (Remarks to the Author):

1: The explanation why the authors did not examine the mRNA level after RNAi knocking down does not make sense to me. In some cases, the mRNA level may not correlate with proteins. However, the mRNA level of the targeted protein should decrease following RNAi KD, which is critical for validating the efficiency of RNAi.

2: The authors argued against the necessity of using a conditional KO method to validate the specificity of their home-made CaMKII antibody, even though it is a gold standard for confirming a newly made antibody. Instead, they used a non-validated RNAi to test the specificity of the antibody. Suspiciously, the representative image has a weird molecular weight shift for both CaMKII and Tubulin in the group of RNAi. Moreover, even based on this representative image, the decrease in the presence of RNAi is less than 50%, even less than 30% calculated based on this representative image... In addition, there is no control RNAi for a fair comparing, and there is no error bars for judging how reliable this result is. Given that this is a key question that I have asked for many times and given that their CaMKII staining decreased strangely when overexpressed mutated CaMKII in FigS1C, I am not convinced that their newly CaMKII antibody is specific.

3: In their previous reply, they argued that the expression of DLG/PSD-95 is correlated with CaMKII, which is simply wrong as I have mentioned. I wanted to use this as an example to explain why it is important to validate their antibody. It is good to see that their antibody may not cross with DLG (still a problem as their antibody has a clear band between 100-130, which they should show whether that band is affected by their RNAi), but this evidence nevertheless cannot be used to prove their CaMKII antibody is specific.

4: The authors now agree that calcium signaling can modulate CaMKII activity, which is strongly supported by the BAPTA-AM experiments. When PHP is induced, postsynaptic calcium reduces, which in turn decreases CaMKII activity. If this is the case, what is the point to show that in the absence of calcium change, loss of GluRIIA can maintain pCaMKII? In the abstract and throughout the manuscript, the authors tried to convey the idea such as: "Thus, the physical loss of the GluRIIA tail is sensed, rather than reduced Ca²⁺ signaling, to enable retrograde PHP signaling, highlighting a unique, Ca²⁺-independent control mechanism for CaMKII in homeostatic plasticity.", which is very misleading. Indeed, the chance that there is no calcium decrease, and only the loss of GluRIIA never occurs during PHP. Thus, the view, as written in the abstract, "For over 20 years, it was hypothesized that diminished Ca²⁺ influx through postsynaptic GluRs reduces CaMKII activity to enable retrograde PHP signaling. is still correct, which diminishes the novelty what they authors want to claim.

5: As I have mentioned previously, the pCaMKII can not simply be used to reflect the activation of CaMKII. The kinase activity is very different with the autophosphorylation state of CaMKII. Also, the calcium influx near the channel, but not the total calcium change, is important for CaMKII activation. In some conditions, people can take pCaMKII at Thr-286/287 as an indicator of CaMKII activation. However, in the case of this study, given that both the phosphorylation and the activation of CaMKII is dynamic, and given that the authors did not control the precise parameters for examining the dynamic change of pCaMKII, it is not convincing that GluRIIA can maintain the activation of CaMKII in the absence of calcium without showing an effector of the CaMKII activation in their manipulation.

6: The cited literatures and the statement are quite misleading and even wrong. For example, they said in the introduction: "...CaMKII is displaced which enables autophosphorylation of a key Thr residue (T287 in *Drosophila*; T286 in mammals)". In mammals, CaMKII can be auto-phosphorylated at Thr at 286 for the alpha isoform, and can be auto-phosphorylated at Thr at 287 for the other isoforms. For another example, the authors wrote in the discussion: "In contrast to the overwhelming evidence for CaMKII having crucial functions in Hebbian functional and structural plasticity at synapses, however, it is less clear to what extent CaMKII operates in either homeostatic plasticity or retrograde signaling at mammalian synapses.". This is also misleading, as the role of CaMKII in mammalian homeostasis has been well studied (Shin Lee et al *Nature Neuroscience* 2012; Thiagarajan Tsien et al., *Neuron* 2002). Of course, they obviously overstate their findings: "Therefore, homeostatic plasticity at the NMJ utilizes a novel and unique mechanism to regulate CaMKII activity, distinct from the well-known paradigm illustrated by Hebbian plasticity in the mammalian brain.", as calcium change still plays a critical role, as I discussed earlier.

Reviewer #3 (Remarks to the Author):

In this revision, the authors addressed the concerns raised by Reviewer2 with new data in Figure S2 and S6 and some adjustments in the main text. The new data demonstrated the efficacy of the transgenic CaMKII RNAi line using western blots. In addition, they showed that the two NMDARs are not components of the postsynaptic apparatus in larval NMJ, which is rather informative.

It appears that Reviewer 2 is not fully satisfied with these responses on three points:

1. How they validated the efficiency of the RNAi (#1, 2).
2. The language used to describe the role of Ca²⁺ in regulating CaMKII activity (#4) and the overall significance of the study (#6),
3. Whether pCaMKII reflects the activation of CaMKII (#5).

Although I disagree with point 2 and 3 made by Reviewer 2, a small modification in Supplementary figure 2 would improve the quality of the data and help further address point 1. Currently the bar graph of the western blot quantification only showed two genotypes and no error bars. A new quantification with all three genotypes and error bars will be helpful.

With the minor exception of Figure S2, I believe the concerns raised by Reviewer 2 have been adequately addressed, and I support the acceptance of the manuscript for publication in its present form.

RESPONSE TO REVIEWS

We thank the Reviewers for their continued time and efforts in evaluating our manuscript and for providing further constructive comments and criticisms. We apologize for the delay in submitting our revised manuscript; we had to wait for a new gel documentation system and a qPCR system to be repaired before we could complete the final experiments requested.

There remained a few persistent concerns expressed by Reviewer 2 about the specificity of the CaMKII antibodies we generated in this study. As detailed below, we now include new experiments focused on assessing CaMKII mRNA levels and improved immunoblot analysis. In short, these experiments demonstrate:

1) qPCR: We performed qPCR analysis of CaMKII mRNA levels in wild type and CaMKII-RNAi body wall tissue using previously validated primers (see Materials and Methods). We find an ~60% reduction in CaMKII mRNA levels following CaMKII knock-down, congruent with the level of protein reduction observed at the NMJ (Supplementary Figure 1). This data is now included in a new **Supplementary Figure 2A**.

2) Immunoblot analysis: We have included updated immunoblot images of total CaMKII protein levels in wild type, CaMKII-RNAi, and CaMKII-OE animals. In addition, we have included additional immunoblot data and present standard error to the graph showing quantification. All this data is now presented in a new **Supplementary Figure 2B-E**.

This new supplementary figure is included along with additional revisions in response to further questions and concerns raised by Reviewer 2.

Together, these new experiments further establish the specificity of the CaMKII antibodies generated in this study.

RESPONSE TO REVIEWER 2

Reviewer 2 raises a handful of persistent concerns regarding the specificity of the anti-CaMKII antibodies generated in this study.

We detail below our specific responses to the concerns raised by Reviewer 2 with new experimental data and revisions to the text. However, before we respond to the specific comments, would like to confront the central and persistent criticism that Reviewer 2 has continued to raise regarding the specificity of the new anti-CaMKII antibodies we generated in our study. We believe these concerns to be unreasonable for the following reasons:

First, none of the central conclusions in our manuscript depend on the specificity of the anti-CaMKII antibodies. We use an abundance of independent molecular, genetic, electrophysiological, and Ca²⁺ imaging approaches presented in Figures 2, 3, 4, 6, and 8 to bolster the central findings of our manuscript:

A) Diminished postsynaptic Ca²⁺ is not sufficient to induce PHP signaling: This key conclusion depends on Ca²⁺ imaging, the new *GluRIIA*^{Q615R} alleles, PV overexpression, and related electrophysiology. No experiments related to the anti-CaMKII antibodies are necessary to validate these conclusions.

B) The GluRIIA C-tail constitutively suppresses retrograde PHP signaling: Here we use CRISPR/Cas9 approaches and molecular genetics to swap the GluRIIA and GluRIIB C-tails, combined with electrophysiology, to demonstrate that the GluRIIA C-tail exerts a constitutive suppression of retrograde PHP signaling.

Second, there is overwhelming evidence from multiple independent approaches to demonstrate the specificity of the CaMKII antibodies generated in our study.

A) Both antibodies were raised using a short peptide highly specific to the *Drosophila* CaMKII amino acid sequence. The pCaMKII antibody was also affinity purified to ensure specificity. Both antibodies clearly labeled distinct structures in the postsynaptic density (Figure 1), a subcellular compartment where *Drosophila* CaMKII has been shown to localize and function based on previous studies from a variety of independent research groups (Koh et al., 1999; Chen and Featherston, 2005; Hodge et al., 2006). The CaMKII antibody clearly recognized a major band on immunoblots that corresponds to the expected mass of CaMKII, which was reduced in CaMKII-RNAi and enhanced in CaMKII-OE (Supplementary Figure 2).

B) Immunostaining levels from both antibodies decrease after CaMKII-RNAi knock-down and increase after CaMKII overexpression. Furthermore, overexpression of two previously validated CaMKII inhibitory peptides selectively reduces pCaMKII levels, but not total CaMKII, as expected (Supplementary Figure 1).

C) Finally, we demonstrate an intimate and bi-directional relationship between pCaMKII immunofluorescence levels and the presence of the GluRIIA C-tail (Figures 1, 5, 6, 7, 8). This is consistent with the previously observed reduction in pCaMKII levels in *GluRIIA* mutants at the fly NMJ using a completely independent, well-established commercial mammalian antibody recognizing pCaMKII found in previous publications and by independent research groups (Hodge et al., 2006; Goel et al., 2017; Newman et al., 2017; Li et al., 2018). Therefore, there is clear and independent precedence for our findings with the *Drosophila*-specific antibodies we generated.

Respectfully, we do not see a coherent alternative possibility that explains the overwhelming evidence detailed above about the CaMKII antibodies that would justify the ongoing concerns raised by the reviewer that they are not specific. Indeed, the data listed above, from a variety of independent approaches (genetics, cell biology, biochemistry, electrophysiology) all support the specificity of our antibodies. Perhaps Reviewer 2 can better articulate how non-specific antibodies, that do not recognize *Drosophila* CaMKII, would still result in all the findings discussed above and presented in our manuscript. It is not clear to us what the alternative possibility is from Reviewer 2's perspective that appears to concern him/her/them so deeply.

Finally, we would also like to point out that we completed major experiments suggested by Reviewer 2 in the first revision focused on the role of synaptic activity and Ca²⁺ levels in CaMKII regulation. In the second revision, we then completed new additional experiments in good faith requested by Reviewer 2 regarding NMDAR function and CaMKII immunoblots, requests that were not brought up in the initial review. Now with the additional experiments urged by Reviewer 2 detailed below, we hope we have allayed the persistent concerns raised by Reviewer 2.

Reviewer 2 states: *The explanation why the authors did not examine the mRNA level after RNAi knocking down does not make sense to me. In some cases, the mRNA level may not correlate with proteins. However, the mRNA level of the targeted protein should decrease following RNAi KD, which is critical for validating the efficiency of RNAi.*

To characterize the knock-down efficiency of the CaMKII-RNAi, we have now performed qPCR analysis requested by Reviewer 2. Indeed, this analysis revealed an ~60% knock-down of CaMKII mRNA, which correlates well with the observed protein levels as assessed by immunostaining and by Western blot. This new data is now shown in a new **Supplementary Figure 2** and discussed in the revised manuscript (**Lines 141-142**).

Reviewer 2 states: *The authors argued against the necessity of using a conditional KO method to validate the specificity of their home-made CaMKII antibody, even though it is a gold standard for confirming a newly made antibody. Instead, they used a non-validated RNAi to test the specificity of the antibody. Suspiciously, the representative image has a weird molecular weight shift for both CaMKII and Tubulin in the group of RNAi. Moreover, even based on this representative image, the decrease in the presence of RNAi is less than 50%, even less than 30% calculated based on this representative image... In addition, there is no control RNAi for a fair comparing, and there is no error bars for judging how reliable this result is. Given that this is a key question that I have asked for many times and given that their CaMKII staining decreased strangely when overexpressed mutated CaMKII in FigS1C, I am not convinced that their newly CaMKII antibody is specific.*

Tissue-specific KO approaches are just emerging in *Drosophila* and still have technical limitations compared to more established approaches. In particular, there is variability in the extent of protein reduction using Cas9-based KO, influenced by maternal contribution, mRNA/protein perdurance, and animal-animal variability in the efficiency of CRISPR-mediated KO. Further complicating such an approach in larval muscle is the fact that muscles are multi-nucleated cells formed by fusion of multiple precursor cells. Finally, if CaMKII levels were reduced sufficiently, the muscles and larvae would likely fail to develop given the essential functions of *Drosophila* CaMKII. Thus, it is clear that conditional CaMKII knock-out in larval muscle tissue using newly developed CRISPR TRiM technology (Mef2-Cas9 + CaMKII-gRNA) does not offer a superior approach over the other strategies we have already used. We previously showed the Reviewer preliminary results of CaMKII-KO in muscle, which produces a significant ~50% reduction in total CaMKII immunostaining levels at the NMJ, comparable to the RNAi approach (see previous **Reviewer Figure 1**).

In response to the Reviewer's additional comments regarding the Western blots, we have now performed additional CaMKII immunoblot analysis in WT, CaMKII-RNAi, and CaMKII overexpression. CaMKII protein levels are significantly reduced following RNAi knockdown and significantly increased following overexpression, indicating that the antibody is specific and sensitive. This new data is now shown in a new **Supplementary Figure 2**.

3. Reviewer 2 states: *In their previous reply, they argued that the expression of DLG/PSD-95 is correlated with CaMKII, which is simply wrong as I have mentioned. I wanted to use this as an example to explain why it is important to validate their antibody. It is good to see that their antibody may not cross with DLG (still a problem as their antibody has a clear band between 100-130, which they should show whether that band is affected by their RNAi), but this evidence nevertheless cannot be used to prove their CaMKII antibody is specific.*

We are not clear to us what point the Reviewer is making here. We mentioned a relationship between DLG and CaMKII in vivo at PSDs in *Drosophila* and in other systems in our previous revision, but never asserted that CaMKII and DLG are directly correlated in expression. Importantly, *we demonstrated that when postsynaptic DLG is virtually eliminated by RNAi (95% reduction), no change in the CaMKII/pCaMKII signal is observed*, which is very strong evidence that the CaMKII antibodies do not cross react with DLG. The only known interaction between *Drosophila* CaMKII and DLG is through

over-phosphorylation of DLG by CaMKII^{T287D}, which impairs its trafficking to the NMJ (Koh et al., 1999), consistent with our data (Supplementary Figure 1C, rightmost panel).

4. Reviewer 2 states: *The authors now agree that calcium signaling can modulate CaMKII activity, which is strongly supported by the BAPTA-AM experiments. When PHP is induced, postsynaptic calcium reduces, which in turn decreases CaMKII activity. If this is the case, what is the point to show that in the absence of calcium change, loss of GluRIIA can maintain pCaMKII? In the abstract and throughout the manuscript, the authors tried to convey the idea such as: “Thus, the physical loss of the GluRIIA tail is sensed, rather than reduced Ca²⁺ signaling, to enable retrograde PHP signaling, highlighting a unique, Ca²⁺-independent control mechanism for CaMKII in homeostatic plasticity.”, which is very misleading. Indeed, the chance that there is no calcium decrease, and only the loss of GluRIIA never occurs during PHP. Thus, the view, as written in the abstract, “For over 20 years, it was hypothesized that diminished Ca²⁺ influx through postsynaptic GluRs reduces CaMKII activity to enable retrograde PHP signaling. is still correct, which diminishes the novelty what they authors want to claim.*

There are many inaccurate statements summarizing the conclusions in our manuscript put forth in the comments by Reviewer 2 above. If these errors were due to our own unclear writing in the manuscript, we apologize and have done our best to ensure our conclusion are accurate with our data. First, we never disagreed that Ca²⁺ signaling can modulate CaMKII activity, this is very well established in *Drosophila* and used to control CaMKII functionality in a variety of processes in the central nervous system (Jin et al., 1998; Thornquist et al., 2020). Put simply, there is *no question Ca²⁺ can modulate CaMKII function*, in *Drosophila* and in mammalian systems, and nowhere in our manuscript did we ever state otherwise.

However, in the context of retrograde PHP signaling at the *Drosophila* NMJ, what we demonstrate is a particularly unique, Ca²⁺-independent mode of regulating CaMKII activity. The Reviewer is correct that in the absence of *GluRIIA*, PHP is induced, and postsynaptic Ca²⁺ is also reduced. It was assumed, as the Reviewer also appears to accept, that this reduction in Ca²⁺ played a role in PHP induction and in controlling CaMKII activity. In our manuscript, we directly test this assumption and demonstrate that **1)** Reduced postsynaptic Ca²⁺ does not induce PHP, and **2)** Postsynaptic CaMKII activity, at least at the larval PSD in the context of PHP signaling, does not change when synaptic activity is silenced (Fig. 3) or when Ca²⁺ is reduced through *GluRIIA* (Fig. 2) or buffered by PV overexpression (Fig. 2). We apologize if these key points were not clear to the Reviewer.

The Reviewer is correct that some manipulations, such as BAPTA-AM, are capable of moderately reducing CaMKII activity at the NMJ (Fig. S5), but the relevance of this experiment to CaMKII activity in the context of PHP signaling is not clear to us. Indeed, we are not clear on what specific points the Reviewer is trying to make here, as Figures 2 and 3 show compelling evidence that reduced Ca²⁺ does NOT reduce CaMKII activity nor is it sufficient to induce PHP (Fig. 4). The key evidence for this claim is the Ca²⁺ impermeable *GluRIIA*^{Q615R} allele, which reduced Ca²⁺ influx into the postsynaptic compartment to the same level as observed in *GluRIIA* mutants (when PHP is normally induced) but has no impact on pCaMKII levels (Fig. 2) nor on PHP expression (Fig. 4). We stand by the important points mentioned by the Reviewer in the abstract and throughout the text, where we have overturned the prominent hypothesis in the field for over 20 years regarding the role of Ca²⁺ in PHP induction.

5. Reviewer 2 states: *As I have mentioned previously, the pCaMKII cannot simply be used to reflect the activation of CaMKII. The kinase activity is very different with the autophosphorylation state of CaMKII. Also, the calcium influx near the channel, but not the total calcium change, is important for CaMKII activation. In some conditions, people can take pCaMKII at Thr-286/287 as an indicator of CaMKII activation. However, in the case of this study,*

given that both the phosphorylation and the activation of CaMKII is dynamic, and given that the authors did not control the precise parameters for examining the dynamic change of pCaMKII, it is not convincing that GluRIIA can maintain the activation of CaMKII in the absence of calcium without showing an effector of the CaMKII activation in their manipulation.

We understand these points made the Reviewer, many of which were shared in the previous review. We have therefore made significant additional revisions to the text to state that we cannot rule out the possibility that changes in CaMKII activity may occur that do not correlate with pCaMKII status (**lines 90-103**).

6. Reviewer 2 states: *The cited literatures and the statement are quite misleading and even wrong. For example, they said in the introduction: “...CaMKII is displaced which enables autophosphorylation of a key Thr residue (T287 in Drosophila; T286 in mammals)”. In mammals, CaMKII can be auto-phosphorylated at Thr at 286 for the alpha isoform, and can be auto-phosphorylated at Thr at 287 for the other isoforms. For another example, the authors wrote in the discussion: “In contrast to the overwhelming evidence for CaMKII having crucial functions in Hebbian functional and structural plasticity at synapses, however, it is less clear to what extent CaMKII operates in either homeostatic plasticity or retrograde signaling at mammalian synapses.”. This is also misleading, as the role of CaMKII in mammalian homeostasis has been well studied (Shin Lee et al Nature Neuroscience 2012; Thiagarajan Tsien et al., Neuron 2002). Of course, they obviously overstate their findings: “Therefore, homeostatic plasticity at the NMJ utilizes a novel and unique mechanism to regulate CaMKII activity, distinct from the well-known paradigm illustrated by Hebbian plasticity in the mammalian brain.”, as calcium change still plays a critical role, as I discussed earlier.*

The Reviewer conflates the term “homeostatic plasticity” which can apply to myriad processes – Ca²⁺ channels at AZs, circuit modulation, etc, with “homeostatic synaptic plasticity”, which is much narrower and typically applies only to postsynaptic receptor scaling or presynaptic homeostatic potentiation or depression (Davis and Müller, 2015). To avoid other readers from conflating these terms, we have revised the text to make clear we are referring to specific, defined forms of homeostatic synaptic plasticity (**lines 421-22**) and cited relevant reviews by Gina Turrigiano, Yuki Goda, and Graeme Davis. We have clarified the language surrounding the connection between pCaMKII levels and overall CaMKII activity in the revised manuscript (**Lines 90-104; 113-114; 168-169; 176-177; 226-227; 244; 399**).

RESPONSE TO REVIEWER 3

Reviewer 3 states: *In this revision, the authors addressed the concerns raised by Reviewer2 with new data in Figure S2 and S6 and some adjustments in the main text. The new data demonstrated the efficacy of the transgenic CaMKII RNAi line using western blots. In addition, they showed that the two NMDARs are not components of the postsynaptic apparatus in larval NMJ, which is rather informative.*

We are grateful to the Reviewer for recognizing the efficacy of the transgenic CaMKII-RNAi line using westerns blots as well as the experiments regarding the two NMDARs detailed in Figures S2 and S6 presented in the previous revision.

It appears that Reviewer 2 is not fully satisfied with these responses on three points:

1. How they validated the efficiency of the RNAi (#1, 2).

2. The language used to describe the role of Ca²⁺ in regulating CaMKII activity (#4) and the overall significance of the study (#6),

3. Whether pCaMKII reflects the activation of CaMKII (#5).

Although I disagree with point 2 and 3 made by Reviewer 2, a small modification in Supplementary figure 2 would improve the quality of the data and help further address point 1. Currently the bar graph of the western blot quantification only showed two genotypes and no error bars. A new quantification with all three genotypes and error bars will be helpful.

We agree with these reasonable requests by Reviewer 3 and now present quantification of Western blots with all three genotypes, including proper error bars. CaMKII protein levels are significantly reduced following RNAi knockdown and significantly increased following overexpression, further supporting the specificity of the CaMKII antibody. This new data is now presented in the new **Supplementary Figure 2.**

With the minor exception of Figure S2, I believe the concerns raised by Reviewer 2 have been adequately addressed, and I support the acceptance of the manuscript for publication in its present form.

We thank the Reviewer for their time and feedback.

REFERENCES

- Chen, K., and Featherston, D. E. (2005). Discs-large (DLG) is clustered by presynaptic innervation and regulates postsynaptic glutamate receptor subunit composition in *Drosophila*. *BMC Biol* 3, 1–13. doi: 10.1186/1741-7007-3-1/FIGURES/5.
- Davis, G. W., and Müller, M. (2015). Homeostatic control of presynaptic neurotransmitter release. *Annu Rev Physiol* 77, 251–270. doi: 10.1146/annurev-physiol-021014-071740.
- Goel, P., Li, X., and Dickman, D. (2017). Disparate Postsynaptic Induction Mechanisms Ultimately Converge to Drive the Retrograde Enhancement of Presynaptic Efficacy. *Cell Rep* 21, 2339–2347. doi: 10.1016/j.celrep.2017.10.116.
- Hodge, J. J. L., Mullasseril, P., and Griffith, L. C. (2006). Activity-Dependent Gating of CaMKII Autonomous Activity by *Drosophila* CASK. *Neuron* 51, 327–337. doi: 10.1016/j.neuron.2006.06.020.
- Jin, P., Griffith, L. C., and Murphey, R. K. (1998). Presynaptic Calcium/Calmodulin-Dependent Protein Kinase II Regulates Habituation of a Simple Reflex in Adult *Drosophila*. *Journal of Neuroscience* 18, 8955–8964. doi: 10.1523/JNEUROSCI.18-21-08955.1998.
- Koh, Y. H., Popova, E., Thomas, U., Griffith, L. C., and Budnik, V. (1999). Regulation of DLG Localization at Synapses by CaMKII-Dependent Phosphorylation. *Cell* 98, 353–363. doi: 10.1016/S0092-8674(00)81964-9.
- Li, X., Goel, P., Chen, C., Angajala, V., Chen, X., and Dickman, D. K. (2018). Synapse-specific and compartmentalized expression of presynaptic homeostatic potentiation. *Elife* 7. doi: 10.7554/eLife.34338.

Newman, Z. L., Hoagland, A., Aghi, K., Worden, K., Levy, S. L., Son, J. H., et al. (2017). Input-Specific Plasticity and Homeostasis at the *Drosophila* Larval Neuromuscular Junction. *Neuron* 93, 1388-1404.e10. doi: 10.1016/j.neuron.2017.02.028.

Thornquist, S. C., Langer, K., Zhang, S. X., Rogulja, D., and Crickmore, M. A. (2020). CaMKII Measures the Passage of Time to Coordinate Behavior and Motivational State. *Neuron* 105, 334-345.e9. doi: 10.1016/J.NEURON.2019.10.018.